# BRUSLEATTACK: A QUERY-EFFICIENT SCORE-BASED BLACK-BOX SPARSE ADVERSARIAL ATTACK

**Viet Quoc Vo, Ehsan Abbasnejad, Damith C. Ranasinghe**
The University of Adelaide
{viet.vo,ehsan.abbasnejad,damith.ranasinghe}@adelaide.edu.au

## ABSTRACT

We study the unique, less-well understood problem of generating *sparse adversarial* samples *simply* by observing the *score-based* replies to *model queries*. Sparse attacks aim to discover a *minimum* number—the $l_0$ bounded—perturbations to model inputs to craft adversarial examples and *misguide* model decisions. But, in contrast to query-based dense attack counterparts against black-box models, constructing sparse adversarial perturbations, even when models serve *confidence score information* to queries in a *score-based* setting, is non-trivial. Because, such an attack leads to: i) an NP-hard problem; and ii) a non-differentiable search space. We develop the BRUSLEATTACK—a *new*, *faster* (more query efficient) Bayesian algorithm for the problem. We conduct extensive attack evaluations including an *attack demonstration* against a Machine Learning as a Service (MLaaS) offering exemplified by **Google Cloud Vision** and robustness testing of adversarial training regimes and a recent defense against black-box attacks. The proposed attack scales to achieve *state-of-the-art attack success rates* and *query efficiency* on standard computer vision tasks such as **ImageNet** across different model architectures. Our artifacts and DIY attack samples are available on GitHub. Importantly, our work facilitates *faster* evaluation of model vulnerabilities and raises our vigilance on the safety, security and reliability of deployed systems.

## 1 INTRODUCTION

We are amidst an increasing prevalence of deep neural networks in real-world systems. So, our ability to understand the safety and security of neural networks is critical to our *trust* in machine intelligence. We have heightened awareness of adversarial attacks (Szegedy et al., 2014)—crafting imperceptible perturbations in inputs to manipulate deep perception systems to produce erroneous decisions. In real-world applications such as machine learning as a service (MLaaS) from Google Cloud Vision or Amazon Rekognition, the model is *hidden* from users. Only, access to model decisions (labels) or confidence *scores* are possible. Thus, crafting adversarial examples in black-box *query-based* interactions with a model is both interesting and practical to consider.

**Why Study *Query*-Based *Sparse* Attacks Under *Score*-Based Responses?** Since confidence scores expose more information compared to model decisions, we can expect fewer queries to elicit effective attacks and, consequently, the potential for developing *attacks at scale* under *score-based* settings. Various similarity measures—$l_p$ norms—are used to quantitatively describe adversarial example perturbations. Particularly, $l_2$ and $l_\infty$ norm is used to quantify **dense** perturbations for attacks. In contrast, $l_0$ norm quantifies *sparse perturbations* aiming to perturb a **tiny** portion of the input. While dense attacks are widely explored, the success of *sparse-attacks*, especially under *score-based* settings, has drawn much less attention and remains less understood (Croce et al., 2022). This leads to our lack of knowledge of model vulnerabilities to sparse perturbation regimes.

**Why are *Score*-Based *Sparse* Attacks Hard?** Constructing sparse perturbations is incredibly difficult as minimizing $l_0$ norm leads to an NP-hard problem (Modas et al., 2019; Dong et al., 2020) and a non-differentiable search space that is mixed (discrete and continuous) (Carlini & Wagner, 2017). Now, for a given $l_0$ constraint or number of pixels, we need to search for both the optimal set of pixels to perturb in a source image and the pixel colors—-floats in [0, 1]. Solutions are harder, if we aim to achieve both *query efficiency* and *high attack success rate* (ASR) for high resolution vision

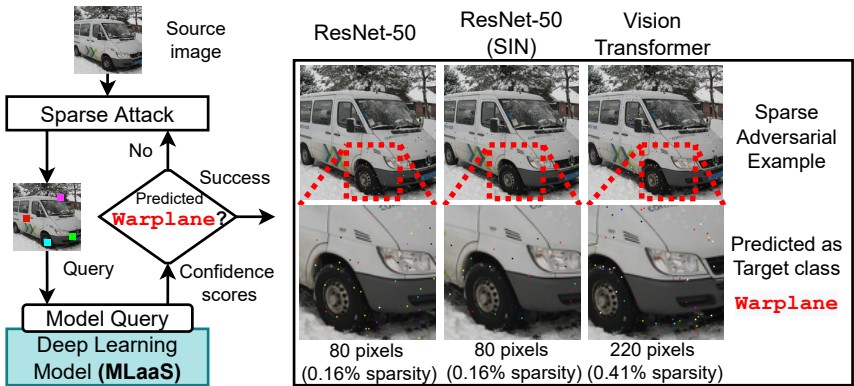

**Figure 1:** *Targeted Attack*. Malicious instances are generated by BRUSLEATTACK with different perturbation budgets against three Deep Learning models on `ImageNet`. An image with ground-truth label `Minibus` is misclassified as a `Warplane`. Interestingly, in contrast to needing *220 pixels* to mislead the Vision Transformer, BRUSLEATTACK requires only *80 perturbed pixels* to fool ResNet-based models (more visuals in **Appendix R**). Evaluation against **Google Cloud Vision** is in **Section** 4.4 and **Appendix** Q.

tasks such as `ImageNet`. The only *scalable* attempt to the challenges, SPARSE-RS (Croce et al., 2022), applies a stochastic search method to seek potential solutions.

**Our Proposed Algorithm.** We consider a *new formulation* to cope with the problem and construct the new search method–BRUSLEATTACK. We propose a search for a sparse adversarial example over an effective, lower dimensional search-space. In contrast to the prior stochastic search and pixel selection method, we guide the search by prior knowledge learned from historical information of pixel manipulations (past experience) and informed selection of pixel level perturbations from our lower dimensional search space to tackle the resulting combinatorial optimization problem.

**Contributions.** Our efforts increase our understanding of *less-well* understood, *hard*, score-based, query attacks to generate *sparse* adversarial examples. Notably, only a few studies exist on the robustness of vision Transformer (ViT) architectures to sparse perturbation regimes. This raises a critical concern over their reliable deployment in applications. Therefore, we investigate the fragility of both CNNs and ViTs against sparse adversarial attacks. Figure 1 demonstrates examples of our attack against models on the `ImageNet` task while we summarize our main contributions below:

- We formulate a new sparse attack—BRUSLEATTACK—in the score-based setting. The algorithm exploits the knowledge of model output scores and our intuitions on: *i)* learning influential pixel information from historical pixel manipulations; and *ii)* informed selection of pixel perturbations based on pixel dissimilarity between our search space prior and a source image to accelerate the search for a *sparse* adversarial example.

- As a *first*, investigate the robustness of ViT and compare its relative robustness with ResNet models on the high-resolution dataset `Imagenet` under score-based sparse settings.

- We demonstrate the significant query efficiency of our algorithm over the state-of-the-art counterpart in different datasets, against various deep learning models as well as defense mechanisms and Google Cloud Vision in terms of ASR & sparsity under 10K query budgets.

## 2    RELATED WORK

**Non-Sparse (Dense) Attacks** ($l_2, l_\infty$)**.** Extensive past works studied dense attacks in white-box (Goodfellow et al., 2014; Madry et al., 2018; Carlini & Wagner, 2017; Dong et al., 2018; Wong et al., 2019; Xu et al., 2020) and black-box settings (Chen et al., 2017; Tu et al., 2019; Liu et al., 2019; Ilyas et al., 2019; Andriushchenko et al., 2020; Shukla et al., 2021; Vo et al., 2022b). Due to non-differentiable, high-dimensional and mixed (continuous & discrete) search space encountered in sparse settings, adopting these methods is non-trivial (see analysis in **Appendix** E). Recent work has explored sparse attacks in white-box settings (Papernot et al., 2016; Modas et al., 2019; Croce & Hein, 2019; Fan et al., 2020; Dong et al., 2020; Zhu et al., 2021). Here we mainly review *sparse* attacks in *black-box* settings but compare with a *white-box sparse attack* for interest in Section 4.2.

***Decision*-based Sparse Attacks** ($l_0$). Only few recent studies, POINTWISE (Schott et al., 2019) and SPARSEEVO (Vo et al., 2022a), have tackled the difficult problem of sparse attacks in decision-based settings. The fundamental difference between decision-based and score-based settings is the output information (labels vs scores) and the ***need*** for a target class image sample in decision-based algorithms. The label information hinders direct optimization from output information. So, decision-based sparse attacks rely on an image from a target class (targeted attacks) and gradient-free methods. This leads to a different set of problem formulations. We study and demonstrate that sparse attacks formulated for *decision*-based settings do not lead to query-efficient attacks in score-based settings in Section 4.2.

***Score*-based Sparse Attacks** ($l_0$). A score-based setting seemingly provides more information than a decision-based setting. But, the first attack formulations (Narodytska & Kasiviswanathan, 2017; Zhao et al., 2019; Croce & Hein, 2019) suffer from prohibitive computational costs (low query efficiency) and do not scale to high-resolution datasets *i.e.* ImageNet. The recent SPARSE-RS random search algorithm in (Croce et al., 2022) reports *the* state-of-the-art, query-efficient, sparse attack and is a significant advance. But, large query budgets are still required to achieve low sparsity on high resolution tasks such as ImageNet in the more difficult targeted attacks.

# 3 PROPOSED METHOD

We focus on exploring adversarial attacks in the context of score-based and sparse settings. First, we present the general problem formulation for sparse adversarial attacks. Let $x \in [0,1]^{c \times w \times h}$ be a normalized source image, where $c$ is the number of channels and $w$, $h$ is the width and height of the image and $y$ is its ground truth label—the *source class*. Let $f(x)$ denote a vector of all class probabilities—softmax scores—from a victim model and $f(r|x)$ denote the probability of class $r$. An attacker aims to search for an adversarial example $\tilde{x} \in [0,1]^{c \times w \times h}$ such that $\tilde{x}$ can be misclassified by the victim model (*untargeted* setting) or classified as a target class $y_{\text{target}}$ (*targeted* setting). Formally, in a targeted setting, for a given $x$, a sparse attack aiming to search for the best adversarial example $x^*$ can be formulated as a constrained combinatorial optimization problem:

$$x^* = \arg\min_{\tilde{x}} L(f(\tilde{x}), y_{\text{target}}) \text{ s.t. } \|x - \tilde{x}\|_0 \leq B, \tag{1}$$

where $\|\|_0$ is the $l_0$ norm denoting the number of perturbed pixels, $B$ denotes a budget of perturbed pixels and $L$ denotes the loss function of the victim model $f$'s predictions. This loss may be different from the training loss and remains unknown to the attacker. In practice, we adopt the loss functions in (Croce et al., 2022), particularly *cross-entropy loss* in targeted settings and *margin loss* in untargeted settings. The problem with Equation 1 is the large search space as we need to search colors, float numbers in $[0,1]$, for perturbing some optimal combination of pixels in the source image $x$.

## 3.1 NEW PROBLEM FORMULATION TO FACILITATE A SOLUTION

Sparse attacks aim to search for the *positions* and *color values* of perturbed pixels; for a normalized image, the color value of each channel of a pixel—RGB color value—can be a float number in $[0,1]$. Consequently, the search space is enormous. Instead of searching in the mixed (discrete and continuous), high-dimensional search space, we consider turning the mixed search space problem into a lower-dimensional, discrete search space problem. Subsequently, we propose a formulation that will aid the development of a new solution to the combinatorial search problem.

**Proposed Lower Dimensional Search Space.** We introduce a simple but effective perturbation scheme. We uniformly sample, at random, a color image $x' \in \{0, 1\}^{c \times w \times h}$—which we call the *synthetic color image*—to define the color of perturbed pixels in the source image $x$. In this manner, each pixel is allowed to attain arbitrary values in $[0,1]$ for each color channel, but the dimensionality of the space is reduced to a discrete space of size $w \times h$. The resulting search space is eight times smaller than the perturbation scheme in SPARSE-RS (Croce et al., 2022) (see an analysis in **Appendix** H). Surprisingly, our proposal is incredibly effective, particularly in high-resolution images such as ImageNet (we provide a comparative analysis with alternatives in **Appendix** I).

**Search Problem Over the Lower Dimensional Space.** Despite the lower-dimensional nature of the search space, a combinatorial search problem persists. As a remedy, we propose changing the problem of finding $\tilde{x}$ to finding a binary matrix $u$ for selecting pixels to perturb in $x$ to construct an

adversarial instance. To that end, we consider choosing a set of pixels in the given image $x$ to be replaced by pixels from the synthetic color image $x' \in \{0,1\}^{c \times w \times h}$. These pixels are determined by a binary matrix $u \in \{0,1\}^{w \times h}$ where $u_{i,j} = 1$ indicates a pixel to be replaced. The adversarial image is then constructed as $\tilde{x} = ux' + (1-u)x$ where $1$ denotes the matrix of all ones with dimensions of $u$, and each element of $u$ corresponds to one pixel of $x$ with $c$ channels. Consequently, manipulating each pixel of $\tilde{x}$ corresponds to manipulating an element in $u$. Therefore, rather than solving Equation 1, we consider the equivalent alternative (proof is shown in **Appendix** G):

$$u^* = \arg\min_{u} \ell(u) \quad \text{s.t.} \ \|u\|_0 \leq B, \tag{2}$$

where $\ell(u) := L(f(ux' + (1-u)x), y_{\text{target}})$. Although the problem in 2 is combinatorial in nature and does not have a polynomial time solution, the formulation facilitates the use of two simple intuitions to iteratively generate better solutions—i.e. sparse adversarial samples.

## 3.2 A BAYESIAN FRAMEWORK FOR THE $l_0$ CONSTRAINED COMBINATORIAL SEARCH

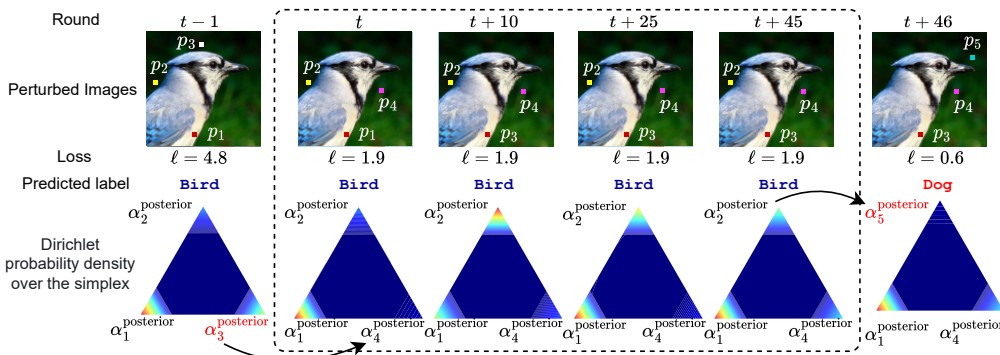

**Figure 2:** A **Sampling** and **Update** illustration. The attack aims to mislead a model into misclassifying a `Bird` image as `Dog`. Assuming that in round $t-1$, an adversarial instance is classified as `Bird` and loss $\ell = 4.8$. We visualize *three elements* of $\alpha^{\text{posterior}}$ for simplicity. Let $\{p_1, p_2, p_3\}$ denote three perturbed pixels with corresponding posterior parameters $\{\alpha_1^{\text{posterior}}, \alpha_2^{\text{posterior}}, \alpha_3^{\text{posterior}}\}$. Assume that in round $t$, two pixels $p_1, p_2$ remain while $p_3$ is replaced by $p_4$ because a loss reduction is observed from 4.8 to 1.9. All $\{\alpha_1^{\text{posterior}}, \alpha_2^{\text{posterior}}, \alpha_3^{\text{posterior}}, \alpha_4^{\text{posterior}}\}$ are updated using Equation 6 but we visualize $\{\alpha_1^{\text{posterior}}, \alpha_2^{\text{posterior}}, \alpha_4^{\text{posterior}}\}$. Since $\alpha_4^{\text{posterior}}$ is new and has never been selected before, it is small in value (and represented using colder colors). From $t$ to $t+45$, while sampling and learning to find a better group of perturbed pixels, $\alpha^{\text{posterior}}$ is updated. Because $p_1$ has a high influence on the model's prediction (represented using warmer colors), it is more likely to remain, while $p_2, p_4$ are more likely to be selected for a replacement due to their lower impact on the model decision. In round $t+46$, pixel $p_2$ is replaced by $p_5$ because a loss reduction is observed from 1.9 to 0.6. Now, the predicted label is flipped from `Bird` to `Dog`.

It is clear that some pixels impart a more significant impact on the model decision than others. As such, given a binary matrix $u$ with a set of selected elements—representing a candidate solution, we can expect some of these elements, if altered, to be more likely to result in an increase in the loss $\ell(u)$. Then, our assumption is that some selected elements must be *hard to manipulate* to reduce the loss, and as such, should be unaltered. Retaining these selected elements is more likely to circumvent a bad solution successfully. *In other words, these selected elements may significantly influence the model's decision and are worth keeping. In contrast to a stochastic search for influential pixels, we consider learning the influence of each element based on the history of pixel manipulations.*

The influence of these elements can be modeled probabilistically, with the more influential elements attaining higher probabilities. To this end, we consider a categorical distribution parameterized by $\theta$, because we aim to select multiple elements and this is equivalent to multiple draws of one of many possible categories. It then follows to consider a Bayesian formulation to learn $\theta$ similar to Abbasnejad et al. (2017). We adopt a general Bayesian framework and ***design the new components and approximations*** needed to learn $\theta$. We can expect a new solution, $u^{(t)}$, generated according to $\theta$ to more likely outweigh the current solution and guide the future candidate solution towards a pixel combination that more effectively minimizing the loss $\ell(u)$. Next, we describe these components and defer the algorithm we have designed, incorporating these components to Section 3.3.

**Prior.** In Bayesian statistics, the conjugate prior distribution of the categorical distribution is the Dirichlet distribution. Thus, we give $\boldsymbol{\theta}$ a prior distribution defined by a Dirichlet distribution with the concentration parameter $\boldsymbol{\alpha}$ as $\mathrm{P}(\boldsymbol{\theta}; \boldsymbol{\alpha}) := \mathrm{Dir}(\boldsymbol{\alpha})$.

**Sampling $\boldsymbol{u}^{(t)}$.** For $t > 0$, given a solution—binary matrix $\boldsymbol{u}^{(t-1)}$—and $\boldsymbol{\theta}^{(t)}$, we aim to: i) select and preserve highly influential selected elements (Equation 3); and ii) draw new elements from unselected elements (Equation 4), conditioned upon $\boldsymbol{u}^{(t-1)} = \mathbf{1}$ and $\boldsymbol{u}^{(t-1)} = \mathbf{0}$, respectively, to jointly yield a new solution $\boldsymbol{u}^{(t)}$ (Equation 5). Concretely, we can express this process as follows:

$$\boldsymbol{v}_1^{(t)} \ldots, \boldsymbol{v}_b^{(t)} \sim \mathrm{Cat}(\boldsymbol{v} \mid \boldsymbol{\theta}^{(t)}, \boldsymbol{u}^{(t-1)} = \mathbf{1}), \tag{3}$$

$$\boldsymbol{q}_1^{(t)}, \ldots, \boldsymbol{q}_{B-b}^{(t)} \sim \mathrm{Cat}(\boldsymbol{q} \mid \boldsymbol{\theta}^{(t)}, \boldsymbol{u}^{(t-1)} = \mathbf{0}), \tag{4}$$

$$\boldsymbol{u}^{(t)} = [\vee_{k=1}^{b} \boldsymbol{v}_k^{(t)}] \vee [\vee_{r=1}^{B-b} \boldsymbol{q}_k^{(t)}]. \tag{5}$$

Here $\boldsymbol{v}_k^{(t)}, \boldsymbol{q}_r^{(t)} \in \{0,1\}^{w \times h}$, $B$ denotes a total number of selected elements (a perturbation budget), $b$ denotes the number of selected elements remaining unchanged, and $\vee$ denotes logical *OR* operator.

**Updating $\boldsymbol{\theta}^{(t)}$ (Using Our Proposed Likelihood).** Finding the exact solution for the underlying parameters $\boldsymbol{\theta}^{(t)}$ of the categorical distribution in Equation 3 and Equation 4 to increase the likelihood of yielding a better solution for $\boldsymbol{u}^{(t)}$ in Equation 5 is often intractable. Our approach is to find an estimate of $\boldsymbol{\theta}^{(t)}$ by obtaining the expectation of the posterior distribution of the parameter, which is learned and updated over time through Bayesian inference. Notably, since the prior distribution of the parameter is a Dirichlet, which is the conjugate prior of the categorical (*i.e.* distribution of $\boldsymbol{u}$), the posterior of the parameter is also Dirichlet. Formally, at each step $t > 0$, updating the posterior and $\boldsymbol{\theta}^{(t)}$ is formulated as follows:

$$\alpha_{i,j}^{\mathrm{posterior}} = \alpha_{i,j}^{\mathrm{prior}} + s_{i,j}^{(t)} \tag{6}$$

$$\mathrm{P}(\boldsymbol{\theta} \mid \boldsymbol{\alpha}, \boldsymbol{u}^{(t-1)}, \ell^{(t-1)}) := \mathrm{Dir}(\boldsymbol{\alpha}^{\mathrm{posterior}}) \tag{7}$$

$$\boldsymbol{\theta}^{(t)} = \mathbb{E}_{\boldsymbol{\theta} \sim \mathrm{P}(\boldsymbol{\theta} \mid \boldsymbol{\alpha}, \boldsymbol{u}^{(t-1)}, \ell^{(t-1)})}[\boldsymbol{\theta}], \tag{8}$$

where $\boldsymbol{\alpha}^{\mathrm{prior}} = \boldsymbol{\alpha}^{(0)}$ is the initial concentration parameter, $\boldsymbol{\alpha}^{\mathrm{posterior}} = \boldsymbol{\alpha}^{(t)}$ denotes the updated concentration parameter (illustration in Figure 2) and $s_{i,j}^{(t)} = ((a_{i,j}^{(t)}) + z)/(n_{i,j}^{(t)} + z)) - 1$. Here, $z$ is a small constant (*i.e.* 0.01) to ensure that the nominator and denominator are always non-zero (this smoothing technique is applied since the nominator and denominator can be zero when "never" manipulated pixels are selected), $a_{i,j}^{(t)}$ is the accumulation of altered pixel $i, j$ (*i.e.* $u_{i,j}^{(t)} = 0$ and $u_{i,j}^{(t-1)} = 1$) when it leads to an increase in the loss, *i.e.* $\ell^{(t)} \geq \ell^{(t-1)}$, and $n_{i,j}^{(t)}$ is the accumulation of selected pixel $i, j$ in the mask $\boldsymbol{u}^{(t)}$. Formally, $a_{i,j}^{(t)}$ and $n_{i,j}^{(t)}$ can be updated as follows:

$$a_{i,j}^{(t)} = \begin{cases} a_{i,j}^{(t-1)} + 1 & \text{if} \quad \ell^t \geq \ell^{(t-1)} \wedge u_{i,j}^{(t)} = 1 \wedge u_{i,j}^{(t-1)} = 0 \\ a_{i,j}^{(t-1)} & \text{otherwise} \end{cases} \tag{9}$$

$$n_{i,j}^{(t)} = \begin{cases} n_{i,j}^{(t-1)} + 1 & \text{if} \quad u_{i,j}^{(t)} = 1 \vee u_{i,j}^{(t-1)} = 1 \\ n_{i,j}^{(t-1)} & \text{otherwise} \end{cases} \tag{10}$$

### 3.3 Sparse Attack Algorithm Formulation With Our Bayesian Framework

Using the Bayesian framework for $l_0$ constrained combinatorial search in Section 3.2, we devise our sparse attack (Algorithm 1) illustrated in Figure 3 and discuss it in detail as follows:

**Initialization** (Algorithm 2). Given a perturbation budget $B$ and a zero-initialized matrix $\boldsymbol{u}$, $N$ first solutions are generated by uniformly altering $B$ elements of $\boldsymbol{u}$ to 1 at random. The initial $\boldsymbol{u}^{(0)}$ is the solution incurring the lowest loss $\ell^{(0)}$. $\boldsymbol{\theta}^{(0)}$ is the expectation of $\boldsymbol{\alpha}^{\mathrm{prior}}$ presented in Appendix M.3.

**Generation** (Algorithm 3). It is necessary here to balance exploration versus exploitation, as in other optimization methods. Initially, to explore the search space, we aim to manipulate a large number of selected elements. When approaching an optimal solution, we aim at exploitation to search for a solution in a region nearby a given solution and thus alter a small number of selected elements. Therefore, we use the combination of power and step decay schedulers to regulate a

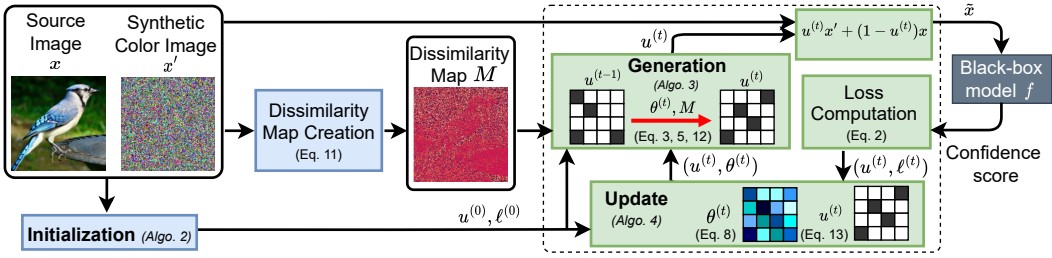

**Figure 3:** BRUSLEATTACK algorithm (Algo. 1). We aim to search for a set of pixels to replace in the source image $x$ by corresponding pixels in a synthetic color image $x'$. In the solution, *binary matrix $\boldsymbol{u}^{(t)}$, white* and black colors denote *replaced* and non-replaced pixels of the source image, respectively. Instead of a stochastic search, we employ our **Bayesian framework** in §**3.2**. **First**, we aim to retain useful elements in the solution $\boldsymbol{u}^{(t)}$ by learning from historical pixel manipulations. For this, we explore and ***learn*** the influence of selected elements by capturing it in the model $\boldsymbol{\theta}$ using our general Bayesian framework in §3.2—darker colors illustrate the higher influence of selected elements (Algo. 4). **Second**, we *generate* new pixel perturbations based on $\boldsymbol{\theta}$ with the ***intuition*** that a larger pixel dissimilarity $M$ between our search space $x'$ and a source image can possibly move the adversarial to the decision boundary faster and accelerate the search (Algo. 3).

---

**Algorithm 1:** BRUSLEATTACK

**Input:** source image $\boldsymbol{x}$, synthetic color image $\boldsymbol{x'}$, source label $y$, target label $y_{\text{target}}$, model $f$
      query limit $T$, scheduler parameters $m_1, m_2$, initial changing rate $\lambda_0$
      perturbation budget $B$, a number of initial samples $N$, concentration parameters $\boldsymbol{\alpha}^{\text{prior}}$

1   Create Dissimilarity Map $\boldsymbol{M}$ using Equation 11
2   $\boldsymbol{u}^{(0)}, \ell^{(0)} \leftarrow \text{INITIALIZATION}(\boldsymbol{x}, \boldsymbol{x'}, y, y_{\text{target}}, N, B, f)$
3   $t \leftarrow 1, \boldsymbol{a}^{(0)} \leftarrow \boldsymbol{0}, \boldsymbol{n}^{(0)} \leftarrow \boldsymbol{u}^{(0)}$
4   Calculate $\boldsymbol{\theta}^{(0)}$ using $\boldsymbol{\alpha}^{\text{prior}}$ and Equation 8
5   **while** $t < T$ *and* $y^{(t)} \neq y_{target}$ **do**
6      $\lambda^{(t)} \leftarrow \lambda_0(t^{m_1} + m_2^t)$
7      $\boldsymbol{u}^{(t)} \leftarrow \text{GENERATION}(\boldsymbol{\theta}^{(t)}, \boldsymbol{M}, \boldsymbol{u}^{(t-1)}, \lambda^{(t)})$
8      $\ell^{(t)} \leftarrow L(f(\boldsymbol{u}^{(t)}\boldsymbol{x'} + (\boldsymbol{1} - \boldsymbol{u}^{(t)})\boldsymbol{x}), y_{\text{target}})$
9      $y^{(t)} \leftarrow \arg\max_r f(r|\boldsymbol{u}^{(t)}\boldsymbol{x'} + (\boldsymbol{1} - \boldsymbol{u}^{(t)})\boldsymbol{x})$
10      $\boldsymbol{u}^{(t)}, \ell^{(t)}, \boldsymbol{\theta}^{(t)}, \boldsymbol{a}^{(t)}, \boldsymbol{n}^{(t)} \leftarrow \text{UPDATE}(\boldsymbol{u}^{(t)}, \ell^{(t)}, \boldsymbol{u}^{(t-1)}, \ell^{(t-1)}, \boldsymbol{a}^{(t)}, \boldsymbol{n}^{(t)})$
11      $t \leftarrow t + 1$
12   **end while**
13   **return** $\boldsymbol{u}^{(t)}$

---

number of selected elements altered in round $t$. This scheduler is formulated as $\lambda_t = \lambda_0(t^{m_1} + m_2^t)$, where $\lambda_0$ is an initial changing rate, $m_1, m_2$ are power and step decay parameters respectively. Concretely, we define a number of selected elements remaining unchanged as $b = \lceil (1 - \lambda_t)B \rceil$.

Given a prior concentration parameter $\boldsymbol{\alpha}^{\text{prior}}$, to generate a new solution in round $t$, we first find $\boldsymbol{\alpha}^{\text{posterior}}$ as in Equation 6 and estimate $\boldsymbol{\theta}^{(t)}$ as in Equation 8. We then generate $\boldsymbol{v}_k^{(t)}$ and $\boldsymbol{q}_r^{(t)}$ as in Equation 3 and Equation 4, respectively. A new solution $\boldsymbol{u}^{(t)}$ can be then formed as in Equation 5. Nonetheless, the naive approach of sampling $\boldsymbol{q}_r^{(t)}$ as in Equation 4 is ineffective and achieves a low performance at low levels of sparsity as shown in Appendix K. When altering unselected elements that are equivalent to replacing non-perturbed pixels in the source image with their corresponding pixels from the synthetic color image, the adversarial instance moves away from the source image by a distance. At a low sparsity level, since a small fraction of unselected elements are altered, the adversarial instance is able to take small steps toward the decision boundary between the source and target class. To mitigate this problem (taking inspiration from (Brunner et al., 2019)) we employ a prior knowledge of the *pixel dissimilarity* between the source image and the synthetic color image. Our intuition is that larger pixel dissimilarities lead to larger steps. As such, it is possible that altering unselected elements with a large pixel dissimilarity moves the adversarial instance to the decision boundary faster and accelerates optimization. The pixel dissimilarity is captured by a dissimilarity

map $\boldsymbol{M}$ as follows:

$$\boldsymbol{M} = \frac{\sum_{c=0}^{2} |\boldsymbol{x}_c - \boldsymbol{x}_c'|}{3}, \tag{11}$$

where $c$ denotes a channel of a pixel. In practice, to incorporate $\boldsymbol{M}$ into the step of sampling $\boldsymbol{q}_r^{(t)}$, Equation 4 is changed to the following:

$$\boldsymbol{q}_1^{(t)}, \dots, \boldsymbol{q}_{B-b}^{(t)} \sim \text{Cat}(\boldsymbol{q} \mid \boldsymbol{\theta}^{(t)} \boldsymbol{M}, \boldsymbol{u}^{(t-1)} = \boldsymbol{0}) \tag{12}$$

**Update** (Algorithm 4). The generated solution $\boldsymbol{u}^{(t)}$ is associated with a loss $\ell^{(t)}$ given by the loss function in Equation 2. This is then used to update $\boldsymbol{\alpha}^{\text{posterior}}$ (Equation 6 and illustration in Figure 2) and the accepted solution as the following:

$$\boldsymbol{u}^{(t)} = \begin{cases} \boldsymbol{u}^{(t)} & \text{if } \ell^{(t)} < \ell^{(t-1)} \\ \boldsymbol{u}^{(t-1)} & \text{otherwise} \end{cases} \tag{13}$$

## 4 EXPERIMENTS AND EVALUATIONS

**Attacks and Datasets.** For a comprehensive evaluation of BRUSLEATTACK, we compose of evaluation sets from CIFAR-10 (Krizhevsky et al.), STL-10 (Coates et al., 2011) and ImageNet (Deng et al., 2009). For CIFAR-10 and STL-10, we select 9,000 and 60,094 different pairs of the source image and target class respectively. For ImageNet, we randomly select 200 *correctly* classified test images evenly distributed among 200 random classes from ImageNet. To reduce the computational burden of the evaluation tasks in the *targeted* setting, five target classes are randomly chosen for each image. For attacks against defended models with Adversarial Training, we randomly select 500 *correctly* classified test images evenly distributed among 500 random classes from ImageNet. We compare with the state-of-the-art SPARSE-RS (Croce et al., 2022).

**Models.** For convolution-based networks, we use models based on a state-of-the-art architecture—ResNet—(He et al., 2016) including ResNet18 achieving 95.28% test accuracy on CIFAR-10, ResNet-9 obtaining 83.5% test accuracy on STL-10, pre-trained ResNet-50 (Marcel & Rodriguez, 2010) with a 76.15% Top-1 test accuracy, pre-trained Stylized ImageNet ResNet-50—ResNet-50 (SIN)—with a 76.72% Top-1 test accuracy (Geirhos et al., 2019) on ImageNet. For the attention-based network, we use a pre-trained ViT-B/16 model achieving 77.91% Top-1 test accuracy (Dosovitskiy et al., 2021). For robust ResNet-50 models [1], we use adversarially pre-trained $l_2 / l_\infty$ models ($l_2$-At and $l_\infty$-AT) (Logan et al., 2019) with 57.9% and 62.42% clean test accuracy respectively.

**Evaluation Metrics.** We define a *sparsity* metric as the number of perturbed pixels divided by the total pixels of an image. To evaluate the performance of an attack, we use *Attack Success Rate* (ASR). A generated perturbation is successful if it can yield an adversarial example with sparsity *below a given sparsity threshold*, then ASR is defined as *the number of successful attacks over the entire evaluation set* at different sparsity thresholds. We measure the *robustness* of a model by the accuracy of that model under an attack at different query limits and sparsity levels.

### 4.1 ATTACK TRANSFORMERS & CONVOLUTIONAL NETS

We carry out comprehensive experiments on ImageNet under the targeted setting to investigate sparse attacks against various Deep Learning models (standard ResNet-50, ResNet-50 (SIN) and ViT). The results for the targeted and untargeted setting are detailed in Appendix B. Additional results on STL-10 and CIFAR-10 are provided in Appendix C and D respectively.

**Convolutional-based Models.** Figure 4a and 4b show that, at sparsity 0.4% ($\approx \frac{200}{224 \times 224}$), BRUSLEATTACK achieves slightly higher ASR than SPARSE-RS while at sparsity 1.0% ($\approx \frac{500}{224 \times 224}$), our attack significantly outweighs SPARSE-RS at different queries. Particularly, from 2K to 6K queries, BRUSLEATTACK obtains about 10% higher ASR than SPARSE-RS. Interestingly, with a small query budget of 6K queries, BRUSLEATTACK to achieve ASR higher than 90%.

**Attention-based Model.** Figure 4c demonstrates that at sparsity of 0.4% BRUSLEATTACK achieves a marginally higher ASR than SPARSE-RS whereas at sparsity of 1.0% our attack demonstrates

---

[1] https://github.com/MadryLab/robustness

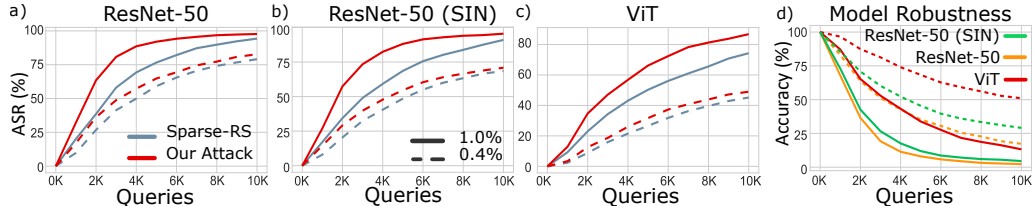

**Figure 4: Targeted setting** on `ImageNet`. a-c) ASR of BRUSLEATTACK and SPARSE-RS against different models at sparsity levels of 0.4% (dashed lines) and 1.0% (solid lines); d) Accuracy of different models against BRUSLEATTACK at sparsity levels (0.4% dash, 1.0% solid; in-between sparsity levels in Appendix B).

significantly better ASR than SPARSE-RS. At 1.0% sparsity and with query budgets above 2K, our method achieves roughly 10 % higher ASR than SPARSE-RS. Overall, our method consistently outperforms the SPARSE-RS in terms of ASR across different query budgets and sparsity levels.

**The Robustness of Transformer versus CNN.** Figure 4d demonstrates the robustness of ResNet-50, ResNet-50 (SIN) and ViT models to adversarially sparse perturbation in the targeted settings. We observe that the performance of all three models degrades as expected. Although ResNet-50 (SIN) is more robust to several types of image corruptions than the standard ResNet-50 by far as shown in (Geirhos et al., 2019), it is as vulnerable as its standard counterpart against sparse adversarial attacks. Interestingly, our results in Figure 4d illustrate that ViT is *much less susceptible* than ResNet family against adversarially sparse perturbation. At the sparsity of 0.4% and 1.0 %, the accuracy of ViT is pragmatically higher than both ResNet models under our attack across different queries. Interestingly, BRUSLEATTACK merely requires a *small query budget of 4K* to degrade the accuracy of both ResNet models to the same accuracy of ViT at 10K queries. These findings can be explained that ViT's receptive field spans over the whole image (Naseer et al., 2021) because some attention heads of ViT in the lower layers pay attention to the entire image (Paul & Chen, 2022). It is thus capable of enhancing relationships between various regions of the image and is harder to be evaded than convolutional-based models if a small subset of pixels is manipulated.

## 4.2 COMPARE WITH PRIOR DECISION-BASED AND $l_0$-ADAPTED ATTACK ALGORITHMS

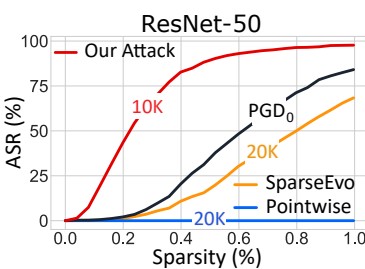

**Figure 5:** Targeted attacks on the `ImageNet` task against ResNet-50. ASR comparisons between BRUSLEATTACK and baselines: i) SPARSEEVO and POINTWISE (SOTA algorithms from ***decision-based*** settings); ii) PGD$_0$ (***whitebox***).

In this section, we compare our method (10K queries) with baselines—SPARSEEVO (Vo et al., 2022a), Pointwise (Schott et al., 2019) (both 20K queries) and PGD$_0$ (Croce & Hein, 2019; Croce et al., 2022) (white-box)—in targeted settings. Figure 5 demonstrates that BRUSLEATTACK significantly outperforms SPARSEEVO and PGD$_0$. For SPARSEEVO and Pointwise, this is expected because decision-based attacks and have only access to the hard label. For PGD$_0$, it is surprised but understandable since in the $l_0$ project step, PGD$_0$ has to identify the minimum number of pixels required for projecting such that the perturbed image remains adversarial but to the best of our knowledge, there is no effective projection method to identify the pixels that can satisfy this projection constraint. *Solving $l_0$ projection problem also lead to another NP-hard problem (Modas et al., 2019; Dong et al., 2020) and hinders the adoption of dense attack algorithms to the $l_0$ constraint.* Moreover, the discrete nature of the $l_0$ ball impedes its amenability to continuous optimization (Croce et al., 2022). Additional results for $l_0$ adapted attacks on `CIFAR-10` are presented in Appendix E.

## 4.3 ATTACK DEFENDED MODELS

BRUSLEATTACK *versus* SPARSE-RS. In this section, we investigate the robustness of sparse attacks (with a budget of 5K queries) against adversarial training-based models using Projected Gradient Descent (PGD) proposed by (Madry et al., 2018)—highly effective defense mechanisms against adversarial attacks (Athalye et al., 2018) and Random Noise Defense (RND) (Qin et al., 2021)—a recent defense method designed for black-box attacks. The robustness of the attacks is measured by

the degraded accuracy of defended models under attacks at different sparsity levels. The stronger an attack is, the lower the accuracy of a defended model is. Table 1 shows that BRUSLEATTACK consistently outweighs SPARSE-RS against different defense methods and different sparsity levels. Additional results on CIFAR-10 is provided in Appendix F.

**Table 1:** Robustness comparison (lower ↓ is stronger) against undefended and defended models employing widely applied adversarial train regimes and the recent RND balckbox attack defence on the ImageNet task. Robustness is measured by the degraded accuracy of models under attacks at different sparsity levels.

| Sparsity | Undefended Model | | $l_\infty$-AT | | $l_2$-AT | | RND | |
|---|---|---|---|---|---|---|---|---|
| | SPARSE-RS | BRUSLEATTACK | SPARSE-RS | BRUSLEATTACK | SPARSE-RS | BRUSLEATTACK | SPARSE-RS | BRUSLEATTACK |
| 0.04% | 33.6% | **24.0**% | 43.8% | **42.2**% | 89.8% | **88.4**% | 90.8% | **85.0**% |
| 0.08% | 13.2% | **6.8**% | 26.8% | **24.4**% | 81.2% | **79.2**% | 82.2% | **72.6**% |
| 0.12% | 7.6% | **2.6**% | 19.0% | **18.4**% | 75.8% | **73.8**% | 73.6% | **61.0**% |
| 0.16% | 5.2% | **1.0**% | 16.6% | **14.8**% | 71.4% | **69.2**% | 64.8% | **51.4**% |
| 0.2% | 4.6% | **1.0**% | 12.2% | **11.8**% | 68.4% | **66.4**% | 56.8% | **42.6**% |

*Undefended and Defended Models.* The results in Table 1 shows the accuracy of undefended versus defended models against sparse attacks across different sparsity levels. In particular, under BRUSLEATTACK and sparsity of 0.2%, the accuracy of ResNet-50 drops to 1% while $l_\infty$-AT model is able to obtain 11.8%. However, $l_2$-AT model and RND strongly resist adversarially sparse perturbation and remains high accuracy around 66.4% and 42.6 % respectively. Therefore, $l_2$-AT model and RND are more robust than $l_\infty$-AT model to defense a model against sparse attacks.

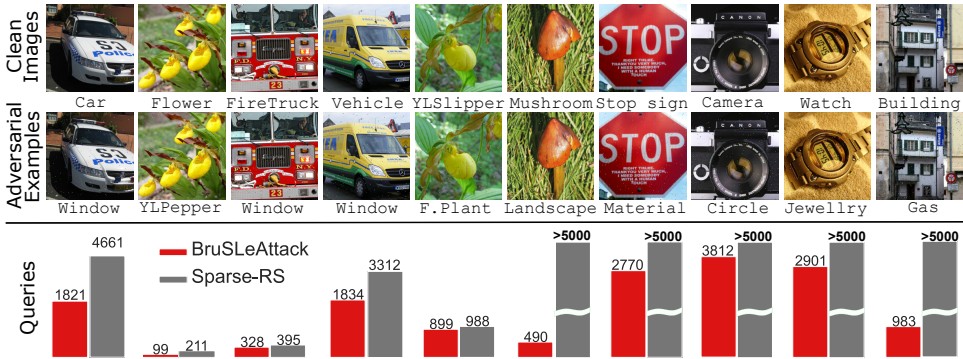

**Figure 6: Demonstration** of sparse attacks against GCV in targeted settings with a budget of 5K queries and sparsity of 0.5% ≈ $\frac{250}{224 \times 224}$. BRUSLEATTACK can yield adversarial examples for all clean images with less queries than SPARSE-RS while SPARSE-RS fails to yield adversarial examples for Mushroom, Camera, Watch, & Building images. Illustration on **GCV API** (online platform) is shown in Appendix Q.

## 4.4 ATTACK DEMONSTRATION AGAINST A REAL-WORLD SYSTEM

To illustrate the applicability and efficacy of BRUSLEATTACK against real-world systems, we attack the Google Cloud Vision (GCV) provided by Google. Attacking GCV is considerably challenging since *1) the classifier returns partial observations of predicted scores with a varied length based on the input and 2) the scores are neither probabilities (softmax scores) nor logits* (Ilyas et al., 2018; Guo et al., 2019). To address these challenges, we employ the *marginal loss* between the top label and the target label and successfully demonstrate our attack against GCV. With a budget of 5K queries and sparsity of 0.5%, BRUSLEATTACK can craft a sparse adversarial example of all given images to mislead GCV whereas SPARSE-RS fails to attack four of them as shown in Figure 6.

## 5 CONCLUSION

In this paper, we propose a novel sparse attack—BRUSLEATTACK. We demonstrate that when attacking different Deep Learning models including undefended and defended models and in different datasets, BRUSLEATTACK consistently achieves better performance than the state-of-the-art method in terms of ASR at different query budgets. Tremendously, in a high-resolution dataset, our comprehensive experiments show that BRUSLEATTACK is remarkably query-efficient and reaches higher ASR than the current state-of-the-art sparse attack in score-based settings.

ACKNOWLEDGEMENTS

This work is supported in part by Google Cloud Research Credits Program and partially supported by the Australian Research Council (DP240103278). The attack is named in honor of Bruce Lee, a childhood hero—ours is a Query-Efficient Score-Based Black-Box Adversarial Attack built upon our proposed Bayesian framework—BRUSLEATTACK.

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

## A    NOTATION TABLE

In this section, we list all notations in Table 2 to help the reader better understand the notations used in this paper.

**Table 2:** Notations used in the paper.

| Notation | Description |
|---|---|
| $\boldsymbol{x}$ | Source image |
| $\tilde{\boldsymbol{x}}$ | Synthetic color image |
| $y$ | Source class |
| $y_{\text{target}}$ | Target class |
| $f(\boldsymbol{x})$ | Softmax scores |
| $L(.)$ or $\ell(.)$ | Loss function |
| $B$ | A budget of perturbed pixels |
| $b$ | A number of selected elements remaining unchanged |
| $\boldsymbol{u}^{(t)}$ | A binary matrix to determine perturbed and unperturbed pixels |
| $\boldsymbol{v}^{(t)}$ | A binary matrix to determine perturbed pixels remaining unchanged |
| $\boldsymbol{q}^{(t)}$ | A binary matrix to determine new pixels to be perturbed |
| $\boldsymbol{\alpha}^{\text{prior}}$ | An initial concentration parameter |
| $\boldsymbol{\alpha}^{\text{posterior}}$ | An updated concentration parameter |
| $\boldsymbol{\theta}$ | Parameter of Categorical distribution |
| $\text{Dir}(\boldsymbol{\alpha})$ | Dirichlet distribution |
| $\text{Cat}(\boldsymbol{\theta})$ | Categorical distribution |
| $\lambda_0$ | An initial changing rate |
| $m_1$ | A power decay parameter |
| $m_2$ | A step decay parameter |
| $\boldsymbol{M}$ | Dissimilarity Map |
| $w,\ h,\ c$ | Width, height and number of channels of an image |

## B    SPARSE ATTACK EVALUATIONS ON IMAGENET

**Table 3:** ASR at different sparsity levels across different queries (higher is better). A comprehensive comparison among different attacks (SPARSE-RS and BRUSLEATTACK) against various Deep Learning models on `ImageNet` in the targeted setting.

| Query | ResNet-50 | | ResNet-50(SIN) | | ViT | |
|---|---|---|---|---|---|---|
| | SPARSE-RS | BRUSLEATTACK | SPARSE-RS | BRUSLEATTACK | SPARSE-RS | BRUSLEATTACK |
| Sparsity = 0.4% | | | | | | |
| 4000 | 49.9% | **57.3**% | 40.5% | **47.8**% | 21.5% | **26.0**% |
| 6000 | 65.5% | **69.4**% | 55.0% | **60.4**% | 31.8% | **37.3**% |
| 8000 | 74.1% | **77.3**% | 63.3% | **66.6**% | 39.6% | **43.9**% |
| 10000 | 79.1% | **82.7**% | 68.5% | **70.9**% | 45.2% | **49.0**% |
| Sparsity = 0.6% | | | | | | |
| 4000 | 59.6% | **75.1**% | 49.7% | **66.2**% | 30.8% | **40.7**% |
| 6000 | 74.0% | **86.3**% | 65.6% | **77.8**% | 43.7% | **52.0**% |
| 8000 | 85.0% | **90.3**% | 77.6% | **83.4**% | 52.2% | **61.0**% |
| 10000 | 90.9% | **93.0**% | 84.3% | **87.0**% | 61.7% | **67.3**% |
| Sparsity = 0.8% | | | | | | |
| 4000 | 65.8% | **84.3**% | 56.3% | **76.7**% | 38.2% | **49.4**% |
| 6000 | 79.2 | **90.6**% | 71.1% | **87.0**% | 50.2% | **63.4**% |
| 8000 | 87.9% | **94.3**% | 81.9% | **91.0**% | 60.0% | **72.2**% |
| 10000 | 93.4% | **96.4**% | 89.6% | **92.4**% | 69.6% | **79.0**% |
| Sparsity = 1.0% | | | | | | |
| 4000 | 69.3% | **88.6**% | 59.2% | **82.4**% | 43.1% | **56.8**% |
| 6000 | 82.1 | **94.2**% | 75.6% | **91.4**% | 56.1% | **72.4**% |
| 8000 | 89.8% | **96.8**% | 83.8% | **94.0**% | 65.6% | **81.3**% |
| 10000 | 94.3% | **97.7**% | 91.0% | **95.5**% | 74.3% | **86.8**% |

**Targeted Settings.** Table 3 shows the detailed ASR results for sparse attacks on high-resolution dataset `ImageNet` in the targeted settings shown in Section 4.1. The results illustrate that the proposed method is consistently better than SPARSE-RS across different sparsity levels from 0.4 % to 1.0 %.

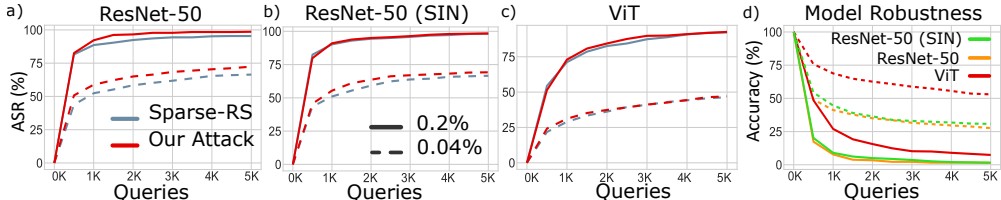

**Figure 7:** a-c) **Untargetted Setting**. ASR versus the number of model queries against different Deep Learning models at sparsity levels (0.4%, 1.0%); d) Accuracy versus the number of model queries for model robustness comparison against BRUSLEATTACK, in the untargeted setting and at sparsity levels ($0.04\% = \frac{40}{224\times224}$, $0.2\% = \frac{100}{224\times224}$).

**Untargeted Settings.** In this section, we verify the performance of sparse attacks against different Deep Learning models including ResNet-50, ResNet-50 (SIN) and ViT models in the untargeted setting up to a 5K query budget. We use an evaluation set of 500 random pairs of an image and a target class to conduct this comprehensive experiment. Our results in Table 4 and Table 7a-c show that BRUSLEATTACK is marginally better than SPARSE-RS across different sparsity levels when attacking against ViT. For ResNet-50 and ResNet-50 (SIN), at lower sparsity or lower query limits, our proposed attack outperforms SPARSE-RS while at higher query budgets or higher sparsity levels, SPARSE-RS is able to obtain slightly lower ASR than our method. In general, BRUSLEATTACK consistently outperforms SPARSE-RS and only needs 1K queries and sparsity of 0.2% (100 pixels) to achieve above 90% ASR against both ResNet-50 and ResNet-50 (SIN).

**Table 4:** ASR at different sparsity levels across different queries (higher is better). A comprehensive comparison among different attacks (SPARSE-RS and BRUSLEATTACK) and various DL models on `ImageNet` in the untargeted setting.

| Query | ResNet-50 | | ResNet-50(SIN) | | ViT | |
|---|---|---|---|---|---|---|
| | SPARSE-RS | BRUSLEATTACK | SPARSE-RS | BRUSLEATTACK | SPARSE-RS | BRUSLEATTACK |
| Sparsity = 0.04% | | | | | | |
| 1000 | 52.4% | **58.8**% | 51.0% | **55.4**% | 29.0% | **31.2**% |
| 2000 | 58.4% | **65.0**% | 59.2% | **63.6**% | 36.2% | **37.4**% |
| 3000 | 61.8% | **68.4**% | 63.8% | **67.0**% | 41.0% | **41.2**% |
| 4000 | 65.4% | **70.4**% | 65.8% | **68.2**% | 44.2% | **44.4**% |
| 5000 | 66.4% | **72.4**% | 66.6% | **69.2**% | 46.4% | **46.7**% |
| Sparsity = 0.08% | | | | | | |
| 1000 | 72.8% | **77.4**% | 73.8% | **75.8**% | 47.2% | **50.6**% |
| 2000 | 81.2% | **86.8**% | 80.4% | **83.4**% | 57.6% | **61.0**% |
| 3000 | 84.6% | **89**% | 84.4% | **87.0**% | 64.2% | **67.8**% |
| 4000 | 85.6% | **90.4**% | 86.6% | **88.2**% | 69.6% | **72.6**% |
| 5000 | 86.8% | **90.8**% | 87.0% | **88.6**% | 72.6% | **74.6**% |
| Sparsity = 0.16% | | | | | | |
| 1000 | 87.0% | **89.4**% | 87.6% | **88.0**% | 64.8% | **68.6**% |
| 2000 | 90.8% | **95.2**% | 92.0% | **94.0**% | 78.4% | **81.4**% |
| 3000 | 93.4 | **96.8**% | 94.8% | **95.6**% | 85.0% | **86.4**% |
| 4000 | 94.4% | **97.6**% | 96.2% | **97.0**% | 87.0% | **89.2**% |
| 5000 | 94.8% | **98.4**% | 96.8% | **97.4**% | 89.8% | **90.0**% |
| Sparsity = 0.2% | | | | | | |
| 1000 | 88.6% | **92.2**% | 90.2% | **91.0**% | 71.2% | **73.0**% |
| 2000 | 92.4% | **96.6**% | 94.4% | **95.0**% | 82.6% | **84.4**% |
| 3000 | 94.4 | **97.8**% | 95.8% | **96.4**% | 87.4% | **89.8**% |
| 4000 | 95.2% | **98.4**% | 97.2% | **98.0**% | 90.8% | **91.0**% |
| 5000 | 95.4% | **98.6**% | 98.2% | **98.4**% | 92.2% | **92.6**% |

**Relative Robustness Comparison among Models.** To compare the relative robustness of different models, we evaluate these models against our attack. Table 4 and Figure 7d confirm our observations

about relative robustness of ResNet-50 (SIN) to the standard ResNet-50 in the targeted setting (presented in Section 4.1). It turns out that ResNet-50 (SIN) is as vulnerable as the standard ResNet-50 even though it is robust against various types of image distortion. Interestingly, ViT is more robust than its convolutional counterparts under sparse attack. Particularly, at sparsity of 0.2% and 2K queries, while the accuracy of both ResNet-50 and ResNet-50 (SIN) is down to about 5%, ViT is still able to remain ASR around 15%.

## C   SPARSE ATTACK EVALUATIONS ON STL10 (TARGETED SETTINGS)

We conduct more extensive experiments on STL-10 in the targeted setting with all correctly classified images of the evaluation set (60,094 sample pairs and image size 96×96). Table 5 provides a comprehensive comparison for different attacks across different sparsity levels ranging from 0.11% (10 pixels) to 0.54% (50 pixels). Particularly, with only 50 pixels, BRUSLEATTACK needs solely 3000 queries to achieve ASR beyond 92% whereas SPARSE-RS only reaches ASR of 89.64%.

**Table 5:** ASR (higher is better) at different sparsity levels in targeted settings. A comprehensive comparison between SPARSE-RS and BRUSLEATTACK against ResNet9 on a full evaluation set from STL-10.

| Methods | Q=1000 | Q=2000 | Q=3000 | Q=4000 | Q=1000 | Q=2000 | Q=3000 | Q=4000 |
|---|---|---|---|---|---|---|---|---|
| | Sparsity = 0.22% | | | | Sparsity = 0.44% | | | |
| SPARSE-RS | 53.82% | 61.65% | 65.84% | 68.0% | 73.34% | 81.47% | 85.24% | 87.49% |
| **BRUSLEATTACK** | **57.69%** | **65.05%** | **68.8%** | **71.22%** | **78.21%** | **85.03%** | **88.31%** | **90.26%** |
| | Sparsity = 0.33% | | | | Sparsity = 0.54% | | | |
| SPARSE-RS | 65.6% | 74.0% | 78.0% | 80.65% | 78.66% | 86.31% | 89.64% | 91.61% |
| **BRUSLEATTACK** | **70.27%** | **77.55%** | **81.16%** | **83.42%** | **83.29%** | **89.78%** | **92.55%** | **94.08%** |

## D   SPARSE ATTACK EVALUATIONS ON CIFAR-10 (TARGETED SETTINGS)

In this section, we conduct extensive experiments in the targeted setting to investigate the robustness of sparse attacks on an evaluation set of 9,000 pairs of an image and a target class from CIFAR-10 (image size 32×32). Sparsity levels range from 1.0% (10 pixels) to 3.9% (40 pixels). Table 6 provides a comprehensive comparison of different attacks in the targeted setting. Particularly, with only 20 pixels (sparsity of 2.0 %), BRUSLEATTACK needs solely 500 queries to achieve ASR beyond 90% whereas SPARSE-RS only reaches ASR of 89.21%. Additionally, with only 300 queries, BRUSLEATTACK is able to reach above 95% of successfully crafting adversarial examples with solely 40 pixels. Overall, our attack consistently outperforms the SPARSE-RS in terms of ASR and this confirms our observations on STL-10 and ImageNet.

**Table 6:** ASR (higher is better) at different sparsity thresholds in the targeted setting. A comprehensive comparison among different attacks (SPARSE-RS and BRUSLEATTACK) against ResNet18 on an evaluation set of 9,000 pairs of an image and a target class from CIFAR-10.

| Methods | Q=100 | Q=200 | Q=300 | Q=400 | Q=500 |
|---|---|---|---|---|---|
| | Sparsity = 1.0% | | | | |
| SPARSE-RS | 36.22% | 50.6% | 58.17 % | 62.59% | 66.26% |
| **BRUSLEATTACK** | **42.32%** | **54.73%** | **61.49%** | **65.33%** | **68.21%** |
| | Sparsity = 2.0% | | | | |
| SPARSE-RS | 60.51% | 76.1% | 83.13% | 86.89% | 89.21% |
| **BRUSLEATTACK** | **66.01%** | **79.19%** | **84.84%** | **88.27%** | **90.24%** |
| | Sparsity = 2.9% | | | | |
| SPARSE-RS | 71.29% | 85.67% | 91.21% | 94.28% | 95.78% |
| **BRUSLEATTACK** | **75.54%** | **88.22%** | **92.91%** | **95.2%** | **96.59%** |
| | Sparsity = 3.9% | | | | |
| SPARSE-RS | 75.91% | 90.21% | 94.78% | 96.97% | 97.98% |
| **BRUSLEATTACK** | **80.44%** | **91.24%** | **95.43%** | **97.4%** | **98.48%** |

# E COMPARING BRUSLEATTACK WITH OTHER ATTACKS ADAPTED FOR SCORE-BASED SPARSE ATTACKS FOR ADDITIONAL BASELINES

## E.1 ADDITIONAL EVALUATIONS WITH DECISION-BASED SPARSE ATTACK METHODS

In this section, we carry out a comprehensive experiment on `CIFAR-10` in the targeted setting (more difficult attack). In our experimental setup, we use an evaluation set of 9000 different pairs of the source image and target classes (1000 images distributed evenly in 10 different classes against 9 target classes) to compare BRUSLEATTACK (500 queries) with SPARSEEVO (2k queries) introduced in (Vo et al., 2022a). We compare ASR of different methods across different sparsity thresholds. The results in Figure 8 demonstrate that our attack significantly outperforms SparseEvo. This is expected because SparseEvo is a decision-based attack and has only access to predicted labels.

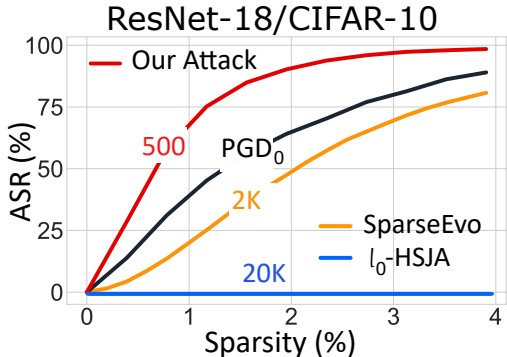

**Figure 8: Targeted attacks** on `CIFAR-10` against ResNet-18. ASR comparisons between BRUSLEATTACK and baselines i) SPARSE-RS and adapted $l_0$-HSJA (***decision-based*** settings); ii) PGD$_0$ (***whitebox***).

**Alternative Loss.** We acknowledge that Vo et al. (2022a) may point out an alternative fitness function based on output scores by replacing optimizing distortion with optimizing loss. However, they did not evaluate their attack method with an alternative fitness function in score-based setting. Employing this alternative fitness function may not obtain a low sparsity level because minimizing the loss does not surely result in a reduction in the number of pixels. Additionally, the Binary Differential Recombination (BDR) in (Vo et al., 2022a) is designed for optimizing $l_0$ distortion not a loss objective (*i.e.* alters perturbed pixels to non-perturbed pixels which is equivalent to minimizing distortion). Hence, naively adapting SPARSEEVO (Vo et al., 2022a) to score-based settings may not work well.

To demonstrate that, we conduct an experiment on `CIFAR-10` using the same experimental setup (same evaluation set of 9000 image pairs and a query budget of 500) described above.

- First approach, we adapted the attack method in (Vo et al., 2022a) to the score-based setting with an alternative fitness function for minimizing loss based on the output scores. We observed this attack always fails to yield an adversarial example with a sparsity level below 50%.

- Second approach, we adapted SPARSEEVO by employing the alternative fitness function, synthetic color image and slightly modifying BDR. Our results in Table 7 show that the adapted SPARSEEVO can create sparse adversarial examples but is unable to achieve a comparable performance to BRUSLEATTACK.

Overall, even with significant improvements, the sparse attack proposed in (Vo et al., 2022a) with an alternative fitness function does not achieve as good performance as BRUSLEATTACK with a low query budget.

**Clarifying Differences Between BRUSLEATTACK and SPARSEEVO (Decision-Based Sparse Attack).** Vo et al. (2022a) develops an algorithms for a sparse attack but assumes a decision-based setting. We compared agianst the attack method and provided results in Figure 5 in the main

**Table 7:** ASR comparison between our proposal and SPARSEEVO (Alternative Loss) on `CIFAR-10`.

| Sparsity | Our Proposal | SPARSEEVO (Alternative Loss) |
|---|---|---|
| 1.0% | **68.21**% | 54.78% |
| 2.0% | **90.24**% | 68.75% |
| 2.9% | **96.59**% | 74.0% |
| 3.9% | **98.48**% | 78.56% |

article. Although both works aim to propose sparse attacks, key differences exist, as expected; we explain these differences below:

- While both works discuss how they reduce dimensionality (a dimensionality reduction scheme) leading to a reduction in search space from $C \times H \times W$ to $H \times W$, Vo et al. (2022a) neither propose a New Problem Formulation nor give proof of showing the equivalent between the original problem in Equation (1) and the New Problem Formulation in Equation (2) as we did in Section 3.1 and Appendix G.

- Our study and Vo et al. (2022a) propose similar terms binary matrix $u$ versus binary vector $v$ as well as an interpolation between $x$ and $x'$. However, a binary vector $x$ in (Vo et al., 2022a) evolves to reduce the number of 1-bits while a binary matrix $u$ in our study maintains a number of 1-elements during searching for a solution.

- We can find a similar notion of employing a starting image (a pre-selected image from a target class) in (Vo et al., 2022a) or synthetic color image (pre-defined by randomly generating) in our study. However, it is worth noting that applying a synthetic color image to Vo et al. (2022a) does not work in the targeted setting. For instance, to the best of our knowledge, there is no method can generate a synthetic color image that can be classified as a target class so the method in (Vo et al., 2022a) is not able to employ a synthetic color image to inialize a targeted attack. In contrast, employing a starting image as used in (Vo et al., 2022a) does not result in query-efficiency as shown in Table 8, especially at low sparsity levels.

Overall, although the score-based setting is less strict than the decision-based setting, our study is not a simplified version of Vo et al. (2022a).

**Table 8:** A comparison of ASR between our proposal (Synthetic Color Image) and employing a starting image as in (Vo et al., 2022a) on `CIFAR-10`.

| Sparsity | Our Proposal | Use starting image |
|---|---|---|
| 1.0% | **68.21**% | 62.68% |
| 2.0% | **90.24**% | 87.17% |
| 2.9% | **96.59**% | 94.37% |
| 3.9% | **98.48**% | 97.17% |

### E.2 IMPACT OF THE BAYESIAN FRAMEWORK BASED SEARCH (ADAPTED SPARSE-RS USING SYNTHETIC IMAGES)

In this section, we conduct an experiment on `ImageNet` and in targeted settings to compare the performance of our method and adapted SPARSE-RS employing synthetic images. Specifically, we replace the update step in SPARSE-RS by fixing the colors to be changed to the ones in a synthetic image. We employ the same evaluation dataset as discussed in Section 4.2.

The results in Table 9 demonstrate that adapted SPARSE-RS is less query-efficient than BRUSLEAT-TACK and even the original SPARSE-RS. In order words, the adapted SPARSE-RS does not benefit from space reduction by employing synthetic images. A possible reason is that the stochastic pixel selection scheme in SPARSE-RS does not leverage historical information on pixel manipulation to determine high and low-influential pixels for preservation or replacement. Therefore, *solely employing synthetic images without our proposed learning framework based on historical information regarding pixel manipulation is not found to achieve high query efficiency*.

**Table 9:** ASR at different sparsity levels across different query budgets (higher is better). A comprehensive comparison among different attacks (SPARSE-RS and BRUSLEATTACK) against various Deep Learning models on `ImageNet` in the targeted setting.

| Query | SPARSE-RS | SPARSE-RS (Synthetic Images) | BRUSLEATTACK |
|---|---|---|---|
| | | Sparsity = 0.4% | |
| 4000 | 49.9% | 49.2% | **57.3**% |
| 6000 | 65.5% | 63.5% | **69.4**% |
| 8000 | 74.1% | 73.6% | **77.3**% |
| 10000 | 79.1% | 79.3% | **82.7**% |
| | | Sparsity = 0.6% | |
| 4000 | 59.6% | 58.7% | **75.1**% |
| 6000 | 74.0% | 73.8% | **86.3**% |
| 8000 | 85.0% | 85.0% | **90.3**% |
| 10000 | 90.9% | 90.0% | **93.0**% |
| | | Sparsity = 0.8% | |
| 4000 | 65.8% | 62.9% | **84.3**% |
| 6000 | 79.2 | 78.7% | **90.6**% |
| 8000 | 87.9% | 87.7% | **94.3**% |
| 10000 | 93.4% | 92.9% | **96.4**% |
| | | Sparsity = 1.0% | |
| 4000 | 69.3% | 67.3% | **88.6**% |
| 6000 | 82.1 | 81.8% | **94.2**% |
| 8000 | 89.8% | 89.7% | **96.8**% |
| 10000 | 94.3% | 93.8% | **97.7**% |

### E.3    $l_0$ ADAPTATIONS OF DENSE ATTACKS

**Adapted $l_0$ Attacks (White-box).** To place the blackbox attack results into context by using a whitebox baseline and to provide a baseline for blackbox attack adaptations to $l_0$, we explore a strong white-box $l_0$ attack. We used $PGD_0$ (Croce & Hein, 2019)—the attack is adaptation of the well-known PGD (Madry et al., 2018) attack. To this end, we compare BRUSLEATTACK with white-box adapted $l_0$ attack $PGD_0$ using the same evaluation set from `CIFAR-10` as decision-based attacks.

The results in Figure 8 demonstrate that our attack significantly outperforms $PGD_0$ at low sparsity threshold and is comparable to $PGD_0$ at high level of sparsity. Surprisingly, our method outweighs white-box, adapted $l_0$ attack $PGD_0$. It is worth noting that there is no effective projection method to identify the pixels that can satisfy sparse constraint and solving the $l_0$ projection problem also encounters an NP-hard problem. Additionally, the discrete nature of the $l_0$ ball impedes its amenability to continuous optimization (Croce et al., 2022).

**Adapted $l_0$ Attacks (Decision-based, Black-box).** It is interesting to adapt $l_2$ attacks such as HSJA (Chen et al., 2020), QEBA (Li et al., 2020), or CMA-ES (Dong et al., 2020) method for face recognition tasks to $l_0$ attacks. Consequently, we adopted the HSJA method to an $l_0$ constraint algorithm called $l_0$-HSJA to conduct a study. For $l_0$-HSJA, we follow the experiment settings and adapted $l_0$-HSJA in (Vo et al., 2022a) and refer to (Vo et al., 2022a) for more details. Notably, the same approach could be adopted for QEBA (Li et al., 2020). The results in Table 10 below illustrate the average sparsity for 100 randomly selected source images, where each image was used to construct a sparse adversarial sample for the 9 different target classes on `CIFAR-10`—hence we conducted 900 attacks or used 900 source-image-to-target-class pairs. The average sparsity across different query budgets is higher than 90% even up to 20K queries. Therefore, the ASR is always 0% at low levels of sparsity (*i.e.* 4%) (shown in Figure 8). These results confirm the findings in (Vo et al., 2022a) and demonstrate that $l_0$-HSJA (20K queries) is not able to achieve good sparsity (lower is better) when compared with our attack method. Consequently, applying an $l_0$ projection to decision-based dense attacks does not yield a strong sparse attack.

Similar to the problem of $PGD_0$, adapted $l_0$-HSJA has to determine a projection that minimizes $l_0$ (the minimum number of pixels) such that the projected instance is still adversarial. To the best of our knowledge, no method in a decision-based setting is able to effectively determine which

pixels can be selected to be projected such that the perturbed image does not cross the unknown decision boundary of the DNN model. Solving this projection problem may also lead to another NP-hard problem (Modas et al., 2019; Dong et al., 2020) and hinders the adoption of these dense attack algorithms to the $l_0$ constraint. Consequently, any adapted method, such as HSJA or other dense attacks, is not capable of providing an efficient method to solve the combinatorial optimization problem faced in sparse settings.

**Table 10:** Mean sparsity at different queries for a targeted setting. A sparsity comparison between $l_0$-HSJA on a set of 100 image pairs on `CIFAR-10`.

| Queries | 4000 | 8000 | 12000 | 16000 | 20000 |
|---|---|---|---|---|---|
| $l_0$-HSJA | 93.66% | 94.73% | 95.88% | 96.74% | 96.74% |

### E.4 COMPARING BRUSLEATTACK WITH ONE-PIXEL ATTACK

In this section, we conduct an experiment to compare BRUSLEATTACK with the One-Pixel Attack Su et al. (2019). We conduct an experiment with 1000 correctly classified images by ResNet18 on CIFAR10 in untargeted settings (notably the easier attack, compared to targeted settings) using ResNet18 These images are evenly distributed across 10 different classes. We compare ASR between our attack and One-Pixel at different budgets e.g. one, three and five perturbed pixels. For One-Pixel attack[2], we used the default setting with 1000 queries. To be fair, we set the same query limits for our attack. The results in Table 11 show that our attack outperforms the One-Pixel attack across one, three and five perturbed pixels, even under the easier, untargeted attack setting.

**Table 11:** ASR comparison (higher ↑ is stronger) between One-Pixel and BRUSLEATTACK against ResNet18 on `CIFAR-10`.

| Perturbed Pixels | One-Pixel | BRUSLEATTACK |
|---|---|---|
| 1 pixel | 19.5% | **27.9**% |
| 3 pixel | 41.9% | **69.9**% |
| 5 pixel | 62.3% | **86.4**% |

### E.5 BAYESIAN OPTIMIZATION

We are interested in the application of Bayesian Optimization for high-dimensional, mix search space. Recently, (Wan et al., 2021) has introduced CASMOPOLITAN, a Bayesian Optimization for categorical and mixed search spaces, demonstrating that this method is efficient and better than other Bayesian Optimization methods in searching for adversarial examples in score-based settings. Therefore, we study and compare our method with CASMOPOLITAN *in the vision domain and the application of seeking sparse adversarial examples*. We note that:

- CASMOPOLITAN solves problem 1 directly by searching for altered pixel positions and the colors for these pixels. In the meanwhile, our method aims to address problem 2, which is reformulated to reduce the dimensionality and complexity of the search space significantly. In general, CASMOPOLITAN aims to search for both color values and pixel positions, whilst BRUSLEATTACK only seeks pixel locations.

- To handle high dimensional search space in an image task, CASMOPOLITAN employs different downsampling/upsampling techniques. It first downscales the image and searches over a low-dimensional space, manipulates and then upscales the crafted examples. Unlike CASMOPOLITAN, our method–BRUSLEATTACK–does not reduce dimensionality by downsampling the original search space but only seeks pixels in an image (source image) and replaces them with corresponding pixels from a synthetic color image (a fixed and pre-defined image) (see Appendix H for our analysis of dimensionality reduction).

---

[2]https://github.com/Harry24k/adversarial-attacks-pytorch

- CASMOPOLITAN is not designed to learn the impact of pixels on the model decisions but treats all pixels equally, whereas BRUSLEATTACK aims to explore the influence of pixels through the historical information of pixel manipulation.

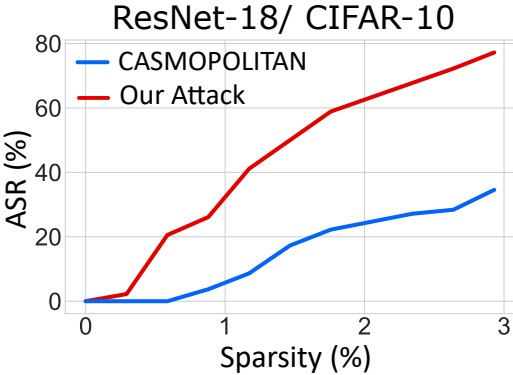

**Figure 9: Targeted attacks** on `CIFAR-10` with a query budget of 250. ASR comparisons between BRUSLEATTACK and CASMOPOLITAN (***Bayesian Optimization***).

We use the code[3] provided in (Wan et al., 2021) and follow their default settings. We evaluate both BRUSLEATTACK and CASMOPOLITAN on an evaluation set of 900 pairs of a source image and a target class from `CIFAR-10` (100 correctly classified images distributed evenly in 10 different classes versus the 9 other classes as target classes for each image) with a query budget of 250. The results in Figure 9 show that BRUSLEATTACK consistently and pragmatically outperforms CAS-MOPOLITAN across different sparsity levels. This is because:

- The mixed search space in the vision domain, particularly in sparse adversarial attacks, is still extremely enormous even if downsampling to a lower dimensional search space. It is because CASMOPOLITAN still needs to search for a color value for each channel of each pixel from a large range of values (see Appendix H for our analysis of dimensionality reduction).
- Searching in a low-dimensional search space and upscaling back to the original search space may not provide an effective way to yield a strong sparse adversarial perturbation. This is because manipulating pixels in a lower dimensional search space may not have the same influence on model decisions as manipulating pixels in the original search space. Additionally, some indirectly altered pixels stemming from upsampling techniques may not greatly impact the model decisions.

### E.6    A DISCUSSION BETWEEN BRUSLEATTACK (ADVERSARIAL ATTACK) AND B3D (BLACK-BOX BACKDOOR DETECTION)

*Natural Evolution Strategies (NES).* A family of black-box optimization methods that learns a search distribution by employing an estimated gradient on its distribution parameters Wierstra et al. (2008); Dong et al. (2021). NES was adopted for score-based dense ($l_2$ and $l_\infty$ norms) attacks in Ilyas et al. (2018) since they mainly adopted a Gaussian distribution for continuous variables. However, solving the problem posed in sparse attacks involving both discrete and continuous variables leads to an NP-hard problem Modas et al. (2019); Dong et al. (2020). Therefore, naively adopting NES for sparse attacks is non-trivial.

The work B3D Dong et al. (2021), in a defense for a data poisoning attack or backdoor attack, proposed an algorithm to reverse-engineer the potential Trojan trigger used to activate the backdoor injected into a model. Although the method is motivated by NES and operates in a score-based setting involving both continuous and discrete variables, as with a sparse attack problem, they are designed for completely different threat models (backdoor attacks with data poisoning versus adversarial attacks). Therefore it is hard to make a direct comparison. However, more qualitatively, there

---

[3]https://github.com/xingchenwan/Casmopolitan

are a number of key differences between our approach and those relevant elements in Dong et al. (2021).

1. Method and Distribution differences: Dong et al. (2021) learns a search distribution determined by its parameters through estimating the gradient on the parameters of this search distribution. In the meantime, our approach is to learn a search distribution through Bayesian learning. While Dong et al. (2021) employed Bernoulli distribution for working with discrete variables, we used Categorical distribution to search discrete variables.

2. Search space (larger vs. smaller): B3D searches for a potential Torjan trigger in an enormous space as it requires to search for pixels' position and color. Our approach reduces the search space and only searches for pixels (pixels' position) to be altered so our search space is significantly lower than the search space used in Dong et al. (2021) if the trigger size is the same as the number of perturbed pixels.

3. Perturbation pattern (square shape vs. any set of pixel distribution): Dong et al. (2021) aims to search for a trigger which usually has a size of $1 \times 1, 2 \times 2$ or $3 \times 3$ so the trigger shape is a small square. In contrast, our attack aims to search for a set of pixels that could be anywhere in an image and the number of pixels could be varied tremendously (determined by desired sparsity). Thus, the combinatorial solutions in a sparse attack problem can be larger than the one in Dong et al. (2021) (even when we equate the trigger size to the number of perturbed pixels).

4. Query efficiency (is a primary objective vs. not an objective): Our approach aims to search for a solution in a query-efficiency manner while it is not clear how efficient the method is to reverse-engineer a trigger.

## F EVALUATIONS AGAINST $l_2, l_\infty$ ROBUST MODELS FROM ROBUSTBENCH AND $l_1$ ROBUST MODELS

**Table 12:** A robustness comparison (lower ↓ is stronger) between SPARSE-RS and BRUSLEATTACK against undefended and defended models employing $l_\infty$, $l_2$ robust models on CIFAR-10. The attack robustness is measured by the degraded accuracy of models under attacks at different sparsity levels.

| Sparsity | Undefended Model | | $l_\infty$-Robust Model | | $l_2$-Robust Model | |
| --- | --- | --- | --- | --- | --- | --- |
| | SPARSE-RS | BRUSLEATTACK | SPARSE-RS | BRUSLEATTACK | SPARSE-RS | BRUSLEATTACK |
| 0.39% | 26.5% | **24.2%** | 65.9% | **65.0%** | 84.7% | **84.2%** |
| 0.78% | 7.8% | **6.4%** | 48.1% | **46.0%** | 70.6% | **68.3%** |
| 1.17% | 2.5% | **2.0%** | 38.1% | **35.1%** | 57.6% | **54.3%** |
| 1.56% | 0.6% | **0.6%** | 28.8% | **26.4%** | 44.4% | **43.8%** |

$l_2, l_\infty$ **Robust Models.** To supplement our demonstration of sparse attacks (BRUSLEATTACK and SPARSE-RS) against defended models on ImageNet in Section 4.3, we consider evaluations against SoTA robust models from RobustBench[4] (Croce et al., 2020) on CIFAR-10. We evaluate the robustness of sparse attacks (BRUSLEATTACK and SPARSE-RS) against the undefended model ResNet-18 and two pre-trained robust models as follows:

- $l_2$ robust model: "Augustin2020Adversarial-34-10-extra". This model is a top-7 robust model (over 20 robust models) in the leaderboard of robustbench.

- $l_\infty$ robust model: "Gowal2021Improving-70-16-ddpm-100m". This model is a top-5 robust model (over 67 robust models) in the leaderboard of robustbench.

We use 1000 samples correctly classified by the pre-trained robust models and evenly distributed across 10 classes on CIFAR-10. We use a query budget of 500. We compare the accuracy of different models (undefended and defended models) under sparse attacks across a range of Sparsity from 0.39% to 1.56%. Notably, defended models are usually evaluated in the untargeted setting to show their robustness. The range of sparsity in the untargeted setting is usually smaller than the range of sparsity used in the targeted setting. Thus, in this experiment, we use a smaller range of sparsity

---
[4]https://github.com/RobustBench/robustbench

than the one we used in the targeted setting. Our results in Table12 show that BRUSLEATTACK outperforms SPARSE-RS when attacking undefended and defended models. The results on CIFAR-10 also confirm our observations on ImageNet.

$l_1$ ***Robust Models.*** We also evaluate our attack method's robustness against $l_1$ robust models. There are two methods AA-I1 Croce & Hein (2021) and Fast-EG-1 Jiang et al. (2023) for training $l_1$ robust models. Although Croce & Hein (2021) and Jiang et al. (2023) illustrated their robustness against $l_1$ attacks, Fast-EG-1 is the current state-of-the-art method (as shown in Jiang et al. (2023)). Therefore, we chose the $l_1$ robust model trained by the Fast-EG-1 method for our experiment. In this experiment, we use 1000 images correctly classified by $l_1$ pre-trained model[5] on CIFAR-10. These images are evenly distributed across ten classes. To keep consistency with previous evaluation, we also use a query budget of 500 and compare the accuracy of the robust model under sparse attacks. The results in Table 13 show that our attack outperforms BRUSLEATTACK across different sparsity levels. Interestingly, $l_1$ robust models are relatively more robust to sparse attacks then other adversarial training regimes in Table 12, this could be because $l_0$ bounded perturbations are enclosed in the $l_1$-norm ball.

**Table 13:** A robustness comparison (lower ↓ is stronger) between SPARSE-RS and BRUSLEATTACK against undefended and defended models employing $l_1$ robust models on CIFAR-10. The attack robustness is measured by the degraded accuracy of models under attacks at different sparsity levels.

| Sparsity | Undefended Model | | $l_1$-Robust Model | |
|---|---|---|---|---|
| | SPARSE-RS | BRUSLEATTACK | SPARSE-RS | BRUSLEATTACK |
| 0.39% | 26.5% | **24.2**% | 86.6% | **85.8**% |
| 0.78% | 7.8% | **6.4**% | 75.8% | **74.8**% |
| 1.17% | 2.5% | **2.0**% | 68.5% | **64.8**% |
| 1.56% | 0.6% | **0.6**% | 59.4% | **55.9**% |

## G    REFORMULATE THE OPTIMIZATION PROBLEM

Solving the problem in Equaion 1 lead to an extremely large search space because of searching colors—float numbers in [0, 1]—for perturbing some pixels. To cope with this problem, we i) reduce the search space by synthesizing a color image $x' \in \{0, 1\}^{c \times w \times h}$—that is used to define the color for perturbed pixels in the source image (see Appendix H), ii) employ a binary matrix $u \in \{0, 1\}^{w \times h}$ to determine positions of perturbed pixels in $x$.

When selecting a pixel, the colors of all three-pixel channels are selected together. Formally, an adversarial instance $\tilde{x}$ can be constructed as follows:

$$\tilde{x} = (1 - u)x + ux' \tag{14}$$

**Proof of The Problem Reformulation.**   Given a source image $x \in [0, 1]^{c \times w \times h}$ and a synthetic color image $x' \in \{0, 1\}^{c \times w \times h}$. From Equation 14, we have the following:

$$\tilde{x} = (1 - u)x + ux'$$
$$\tilde{x} - (1 - u)x = ux'$$
$$u\tilde{x} + (1 - u)\tilde{x} - (1 - u)x = ux'$$
$$(1 - u)(\tilde{x} - x) = u(x' - \tilde{x})$$

We consider two cases for each pixel here:

1. If $u_{i,j} = 0$: then $(1 - u_{i,j})(\tilde{x}_{i,j} - x_{i,j}) = 0$, thus $\tilde{x}_{i,j} = x_{i,j}$
2. If $u_{i,j} = 1$: then $u_{i,j}(x'_{i,j} - \tilde{x}_{i,j}) = 0$, thus $\tilde{x}_{i,j} = x'_{i,j}$

Therefore, manipulating binary vector $u$ is equivalent to manipulating $\tilde{x}$ according to 14. Hence, optimizing $L(f(\tilde{x}), y^*)$ is equivalent to optimizing $L(f((1 - u)x + ux'), y^*)$.

---

[5]https://github.com/IVRL/FastAdvL1

# H  ANALYSIS OF SEARCH SPACE REFORMULATION AND DIMENSIONALITY REDUCTION

Sparse attacks aim to search for the positions and color values of these perturbed pixels. For a normalized image, the color value of each channel of a pixel—RGB color value—can be a float number in $[0, 1]$ so the search space is enormous. The perturbation scheme proposed in (Croce et al., 2022) can be adapted to cope with this problem. This perturbation scheme limits the RGB values to a set $\{0, 1\}$ so a pixel has eight possible color codes $\{000, 001, 010, 011, 100, 101, 110, 111\}$ where each digit of a color code denotes a color value of a channel. This scheme may result in noticeable perturbations but does not alter the semantic content of the input. However, this perturbation scheme still results in a large search space because it grows rapidly with respect to the image size. To obtain a more compact search space, we introduce a simple but effective perturbation scheme. In this scheme, we uniformly sample at random a color image $x' \in \{0, 1\}^{c \times w \times h}$—*synthetic color image*—to define the color of perturbed pixels in the source image $x$. Additionally, we use a binary matrix for selecting some perturbed pixels in $x$ and apply the matrix to $x'$ to extract color for these perturbed pixels as presented in Appendix G. Because $x'$ is generated once in advance for each attack and has the same size as $x$, the search space is eight times smaller than using the perturbation scheme in (Croce et al., 2022). Surprisingly, our elegant proposal is shown to be incredibly effective, particularly in high-resolution images such as `ImageNet`.

**Synthetic color image.**  Our attack method does not optimize but pre-specify a synthetic color image $x'$ by using our proposed random sampling strategy in our algorithm formulation. This synthetic image is generated once, dubbed a one-time synthetic color image, for each attack. We have chosen to generate it once rather than optimizing it because:

- We aim to reduce the dimensionality of the search space to find and adversarial example. Choosing to optimize the color image would lead to a difficult combinatorial optimization problem.
  - Consider what we presented in Section 3.1. To solve the combinatorial optimization problem in Equation 1, we might search a color value for each channel of each pixel–a float number in [0,1] and this search space is enormous. For instance, if we need to perturb $n$ pixels and the color scale is $2^m$, the search space is equivalent to $C_{2^m \times c \times w \times h}^{c \times n}$.
  - To alleviate this problem, we reformulate problem in Equation 1 and proposed a search over the subspace $\{0,1\}^{c \times w \times h}$. However, the size of this search space is still large.
  - To further reduce the search space, we construct a fixed search space—a pre-defined synthetic color image $x' \in \{0,1\}^{c \times w \times h}$ for each attack. The search space is now reduced to $C_{w \times h}^{n}$. It is generated by uniformly selecting the color value for each channel of each pixel from $\{0, 1\}$ at random (as presented in Appendix H and G).
- In addition, a pre-defined synthetic color image $x'$––a fixed search space—benefits our Bayesian algorithm. If keeping optimizing the synthetic color image $x'$, our Bayesian algorithm has to learn and explore a large number of parameters which is equivalent to $C_{2^m \times c \times w \times h}^{c \times n}$ and we might not learn useful information fast enough to make the attack progress.
- Perhaps, most interestingly, our attack demonstrates that a solution for the combinatorial optimization problem in Equation 1 can be found in a pre-defined and fixed subspace.

**Searching for pixels' position and color concurrently.** In general, changing the color of the pixels in searches led to significant increases in query budgets. In our approach, we aim to model the influence of each pixel bearing a specific color, probabilistically, and learn the probability model through the historical information collected from pixel manipulations. So, we chose not to first search for pixels' position and search for their color after knowing the position of pixels but we aim to do both simultaneously. In other words, the solution found by our method is a set of pixels with their specific colors.

# I  DIFFERENT SCHEMES FOR GENERATING SYNTHETIC IMAGES

In this section, we analyze the impact of different schemes including different random distributions, maximizing dissimilarity and low color search space.

**Different random distributions.** Since the synthetic color images are randomly generated, we can leverage Uniform or Gaussian distribution or our method. Because the input must be within $[0, 1]$, we can sample $x'$ from $\mathcal{U}[0, 1]$ or $\mathcal{N}(\mu, \sigma^2)$ where $\mu = 0.5, \sigma = 0.17$. For our method, we uniformly sample at random a color image $x' \in \{0, 1\}^{c \times w \times h}$. In order words, each channel of a pixel receives a binary value 0 or 1. The results in Table 14 show that generating a synthetic color image from Uniform distribution is better than Gaussian distribution but it is worse than our simple method. The experiment illustrates that different schemes of generating the synthetic color image at random have different influences on the performance of BRUSLEATTACK and our proposal outweighs other common approaches across different sparsity levels. Particularly at low query budgets (*i.e.* up to 300 queries) and low perturbation budgets (*i.e.* sparsity up to 3%), our proposal outperforms the other two by a large margin. Therefore, the empirical results show our proposed scheme is more effective in obtaining good performance. Most interestingly, as pointed out by HSJA authors (Chen et al., 2020), the question of how best to select an initialization method or in their case initial target image remains an open-ended question worth investigating.

**Table 14:** Target setting. ASR (higher is better) at different sparsity thresholds in the targeted setting. A comprehensive comparison among different strategies of synthetic color image generation to initialize BRUSLEAT-TACK attack against ResNet18 on CIFAR-10.

| Methods | Q=100 | Q=200 | Q=300 | Q=400 | Q=500 |
|---------|-------|-------|-------|-------|-------|
| Sparsity = 1.0% | | | | | |
| Uniform | 32.18% | 41.68% | 48.09 % | 52.38% | 55.48% |
| Gaussian | 21.29% | 29.87% | 35.0 % | 38.72% | 41.53% |
| **Ours** | **42.32**% | **54.73**% | **61.49**% | **65.33**% | **68.21**% |
| Sparsity = 2.0% | | | | | |
| Uniform | 54.04% | 69.08% | 76.48% | 80.91% | 83.76% |
| Gaussian | 40.02% | 55.17% | 63.2 % | 68.58% | 72.28% |
| **Ours** | **66.01**% | **79.19**% | **84.84**% | **88.27**% | **90.24**% |
| Sparsity = 2.9% | | | | | |
| Uniform | 65.82% | 80.62% | 87.84% | 91.39% | 93.38% |
| Gaussian | 52.4% | 69.91% | 78.42 % | 83.24% | 86.39% |
| **Ours** | **75.54**% | **88.22**% | **92.91**% | **95.2**% | **96.59**% |
| Sparsity = 3.9% | | | | | |
| Uniform | 73.04% | 86.32% | 92.33% | 95.02% | 96.34% |
| Gaussian | 61.0% | 77.26% | 84.88 % | 89.63% | 91.94% |
| **Ours** | **80.44**% | **91.24**% | **95.43**% | **97.4**% | **98.48**% |

**Maximizing dissimilarity.** There may be different ways to implement your suggestion of generating a synthetic color image $x'$ that maximize the dissimilarity between the original image $x$ and $x'$. But to the best our knowledge, no effective method can generate a random color image $x'$ that maximize its dissimilarity with $x$.

Our approach to this suggestion is to find the inverted color values of $x$ by creating an inverted image $x_{invert}$ to explore color values different from $x$. We then find the frequency of these color values (in each R, G, B channel) in $x_{invert}$. Finally, we generate a synthetic color image $x'$ such that the more frequent color values (in R, G, B channels) in $x_{invert}$ will appear more frequently in $x'$. By employing the frequency information of color values in $x$, we can create a synthetic color image $x'$ that is more dissimilar to $x$. In practice, our implementation is described as follow:

- Yield the inverted image $x_{invert}$ = 1 - $x$. Note that $x \in [0, 1]^{c \times w \times h}$

- Create a histogram of pixel colors (for each R, G, B channel) to have their frequency in $x_{invert}$.

- Then we randomly generate a synthetic color image based on the frequency of color values that allows us to maximize the dissimilarity.

The results in Table 15 show that an approach of maximizing the dissimilarity (using frequency information) yields better performance at low sparsity levels as we discussed in Appendix K. However, it does not result in better performance at high levels of sparsity if compared with our proposal.

**Table 15:** ASR comparison between using a synthetic color image uniformly generated at random (our proposal) and maximizing dissimilarity on `CIFAR-10`.

| Sparsity | Our Proposal | Maximizing Dissimilarity |
|---|---|---|
| 1.0% | 68.21% | **70.16**% |
| 2.0% | 90.24% | **90.75**% |
| 2.9% | **96.59**% | 95.78% |
| 3.9% | **98.48**% | 97.85% |

**Low color search space.** Instead of reducing the space from 8 color codes to a fixed random one, we consider choosing between 2-4 random colors. That would allow us to search not only in the position space of the pixels but also in their color space without increasing search space significantly. The results in Table 16 show that expanding color space leads to larger search space. Consequently, this approach may require more queries to search for a solution and results in low ASR, particularly with a small query budget.

**Table 16:** ASR comparison between using a fixed random color search space (our proposal) and two or four random color search space on `CIFAR-10`.

| Sparsity | Our Proposal | Two Random Colors | Four Random Colors |
|---|---|---|---|
| 1.0% | **68.21**% | 60.11% | 57.9% |
| 2.0% | **90.24**% | 78.12% | 78.1% |
| 2.9% | **96.59**% | 85.89% | 90.67% |
| 3.9% | **98.48**% | 91.23% | 95.28% |

## J  BRUSLEATTACK UNDER DIFFERENT RANDOM SEEDS

It is possible that the initial generated by uniformly selecting the color value for each channel of each pixel from $\{0, 1\}$ at random (as presented in Appendix H and Appendix G) could impact performance. We investigate this using Monte Carlo experiments. To analyze if our attack is sensitive to our proposed initialization scheme. We run our attack 10 times with different random seeds for each source image and target class pair. This also generates 10 different synthetic color images ($x'$) for each source image and target class pair. We chose an evaluation set of 1000 source images (evenly distributed across 10 random classes) and used each one and our attack to flip the label to 9 different target classes. So we conducted ($1000 \times 9$ source-image-to-target-class pairs) $\times$ 10 (ten because we generated 10 different for each pair) attacks (90K attacks) against ResNet18 on `CIFAR-10`. We report the min, max, average and standard deviation ASR across the entire evaluation set at different sparsity levels. The results in Table 17 show that our method is invariant to the initialization of $x'$. Therefore, our initialization scheme does not affect the final performance of our attacks reported in the paper. Actually, the more complex task of optimizing $x'$ and devising efficient algorithms to explore the high dimensional search space or the generation of better image synthesizing schemes (initialization schemes) to boost the attack performance leaves interesting works in the future.

**Table 17:** ASR (Min, Mean, Max and Standard Deviation) of our attack methods across the entire evaluation set at different sparsity levels with a query budget of 500, with 10 different random seeds for each attack on `CIFAR-10`.

| Sparsity | ASR (Min) | ASR (Mean) | ASR (Max) | Standard Deviation |
|---|---|---|---|---|
| 1%(10 pixels) | 68.14 % | 68.36 % | 68.66% | 0.35 |
| 2%(20 pixels) | 90.24% | 90.76% | 91.38% | 0.48 |
| 2.9%(30 pixels) | 96.62% | 96.71% | 96.78% | 0.11 |
| 3.9%(40 pixels) | 98.17% | 98.35% | 98.49% | 0.13 |

# K    EFFECTIVENESS OF DISSIMILARITY MAP

In this section, we aim to investigate the impact of employing the dissimilarity map as our prior knowledge.

**On `CIFAR-10`.** Similarly, we conduct another experiment on an evaluation set which is composed of 1000 correctly classified images (from `CIFAR-10`) evenly distributed in 10 classes and 9 target classes per image. However, to reduce the burden of computation when studying hyper-parameters, we use a query budget of 500. The results in Table 18 confirm our observation on `ImageNet`.

**Table 18:** ASR comparison between with and without using Dissimilarity Map on `CIFAR-10`.

| Sparsity | With Dissimilarity Map | Without Dissimilarity Map |
|---|---|---|
| 1.0% | **68.21**% | 67.16% |
| 2.0% | **90.24**% | 89.42% |
| 2.9% | **96.59**% | 95.96% |
| 3.9% | **98.48**% | 97.92% |

**On `ImageNet`.** We conduct a more comprehensive experiment in terms of query budget to show more interesting results. In this experiment, we use the same evaluation set of 500 samples from `ImageNet` used in Section 4 and in the targeted setting. The results in Table 19 show that employing prior knowledge of pixel dissimilarity benefits our attack, particularly at a low percentage of sparsity rather. At a high percentage of sparsity, BRUSLEATTACK adopting prior knowledge only achieves a comparable performance to BRUSLEATTACK without prior knowledge. Notably, at a sparsity of 0.2%, BRUSLEATTACK is slightly worse than SPARSE-RS. Nonetheless, employing prior knowledge of pixel dissimilarity improve the performance of BRUSLEATTACK and makes it consistently outweigh SPARSE-RS.

**Table 19:** ASR at different sparsity thresholds and queries (higher is better) for a targeted setting. A comparison between SPARSE-RS, BRUSLEATTACK (without Dissimilarity Map) and BRUSLEATTACK (with Dissimilarity Map) on an evaluation set of 500 pairs of an image and a target class on `ImageNet`
.

| Methods | Q=2000 | Q=4000 | Q=6000 | Q=8000 | Q=10000 |
|---|---|---|---|---|---|
| Sparsity = 0.2% | | | | | |
| SPARSE-RS | 9.4% | 20.6% | 29.6 % | 33.4% | 38.4% |
| BRUSLEATTACK (without Dissimilarity Map) | 8.8% | 19.6% | 27.4 % | 34.4% | 38.2% |
| **BRUSLEATTACK** | **12**% | **23.6**% | **31.6**% | **36.6**% | **40.4**% |
| Sparsity = 0.4% | | | | | |
| SPARSE-RS | 23.6% | 48.4% | 63.0% | 72.6% | 78.8% |
| BRUSLEATTACK (without Dissimilarity Map) | 30.2% | 53.4% | 64.4% | 73.0% | 78.6% |
| **BRUSLEATTACK** | **33.2**% | **54.2**% | **66.8**% | **76**% | **82.4**% |
| Sparsity = 0.6% | | | | | |
| SPARSE-RS | 29.6% | 57.6% | 73.2% | 85.8% | 92.0% |
| BRUSLEATTACK (without Dissimilarity Map) | 43.6% | 71.6% | 85.0% | 91.8% | 94.6% |
| **BRUSLEATTACK** | **45.4**% | **75.6**% | **87.4**% | **91.8**% | **94.6**% |

# L    HYPER-PARAMETERS, INITIALIZATION AND COMPUTATION RESOURCES

All experiments in this study are performed on two RTX TITAN GPU ($2 \times 24$GB) and four RTX A6000 GPU ($4 \times 48$GB). We summarize all hyper-parameters used for BRUSLEATTACK on the evaluation sets from `CIFAR-10`, `STL-10` and `ImageNet` as shown in Table 20 . Notably, only the initial changing rate $\lambda_0$ is adjusted for different resolution datasets *i.e.* `STL-10` or `ImageNet` ; thus, our method can be easily adopted for different vision tasks. Additionally, to realize an attack, we randomly synthesize a color image $x'$ for each attack. At initialization step, BRUSLEATTACK randomly creates 10 candidate solutions and choose the best.

**Table 20:** Hyper-parameters setting in our experiments

| Parameters | CIFAR-10 | | STL-10 | | ImageNet | |
|---|---|---|---|---|---|---|
| | Untargeted | Targeted | Untargeted | Targeted | Untargeted | Targeted |
| $m_1$ | 0.24 | 0.24 | 0.24 | 0.24 | 0.24 | 0.24 |
| $m_2$ | 0.997 | 0.997 | 0.997 | 0.997 | 0.997 | 0.997 |
| $\lambda_0$ | 0.3 | 0.15 | 0.3 | 0.15 | 0.3 | 0.05 |
| $\alpha^{\text{prior}}$ | **1** | **1** | **1** | **1** | **1** | **1** |

# M    HYPER-PARAMETERS STUDY

In this section, we conduct comprehensive experiments to study the impacts and the choice of hyper-parameters used in our algorithm. The experiments in this section are mainly conducted on CIFAR-10. For $\lambda_0$, we conduct an additional experiment on ImageNet.

## M.1    THE IMPACT OF $m_1, m_2$

In this experiment, we use the same evaluation set on CIFAR-10 mentioned above. To investigate the impact of $m_1$, we set $m_2 = 0.997$ and change $m_1 = 0.2, 0.24, 0.28$. Likewise, we set $m_1 = 0.24$ and change $m_2 = 0.993, 0.997, 0.999$ to study $m_2$. The results in Table 21 show that BRUSLEATTACK achieves the best results with $m_1 = 0.24$ and $m_2 = 0.997$.

**Table 21:** ASR of BRUSLEATTACK with different values of $m_1, m_2$ on CIFAR-10.

| Sparsity | Fixed $m_2 = 0.997$ | | | Fixed $m_1 = 0.24$ | | |
|---|---|---|---|---|---|---|
| | $m_1 = 0.2$ | $m_1 = 0.24$ | $m_1 = 0.28$ | $m_2 = 0.993$ | $m_2 = 0.997$ | $m_2 = 0.999$ |
| 1.0% | 67.32% | **68.21**% | 67.48% | 67.34% | **68.21**% | 67.21% |
| 2.0% | 88.67% | **90.24**% | 88.94% | 89.64% | **90.24**% | 89.12% |
| 2.9% | 95.37% | **96.59**% | 95.54% | 96.25% | **96.59**% | 95.82% |
| 3.9% | 97.24% | **98.48**% | 97.68% | 97.59% | **98.48**% | 96.21% |

## M.2    THE IMPACT OF $\lambda_0$

**On CIFAR-10.** Similarly, we conduct another experiment on the same evaluation set which is composed of 1000 correctly classified images (from CIFAR-10) as described above. We use the same query budget of 500. We use $m_1 = 0.24$ and $m_2 = 0.997$ and change $\lambda_0 = 0.15$ to study the impact of $\lambda_0$. Our results in Table 22 show that BRUSLEATTACK achieves the best results with $\lambda_0 = 0.15$.

**Table 22:** ASR of BRUSLEATTACK with different values of $\lambda_0$ on CIFAR-10.

| Sparsity | $\lambda_0 = 0.1$ | $\lambda_0 = 0.15$ | $\lambda_0 = 0.2$ |
|---|---|---|---|
| 1.0% | 68.05% | **68.21**% | 68.12% |
| 2.0% | 89.38% | **90.24**% | 88.33% |
| 2.9% | 96.15% | **96.59**% | 95.56% |
| 3.9% | 98.16% | **98.48**% | 97.08% |

**On ImageNet.** We use 500 random pairs of an image and a target class from ImageNet in a targeted setting. We carry on a more comprehensive experiment in terms of query budgets. Figure 10 shows that with different initial changing rates $\lambda_0$, BRUSLEATTACK obtains the best results when $\lambda_0$ is small such as 0.03 or 0.05. However, at a small sparsity budget, $\lambda_0 = 0.03$ often achieves lower ASR than $\lambda_0 = 0.05$ as shown in Table 23 because it requires more queries to make changes and move towards a solution. Consequently, $\lambda_0$ should not be too small. If increasing $\lambda_0$, BRUSLEATTACK reaches its highest ASR slower than using small $\lambda_0$. Hence, the initial changing rate has an impact on the overall performance of BRUSLEATTACK.

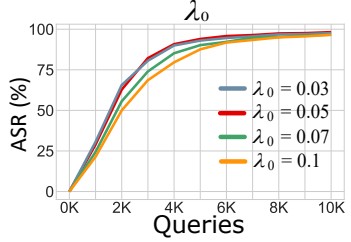

**Figure 10:** ASR versus model queries on `ImageNet`. BRUSLEATTACK against ResNet-50 with sparsity of 1.0 % in a targeted setting to show the impacts of different hyper-parameters on BRUSLEATTACK.

**Table 23:** ASR at different sparsity levels and queries (higher is better) in a targeted setting. A comparison between $\lambda_0 = 0.03$ and $\lambda_0 = 0.05$ on a set of 500 pairs of an image and a target class on `ImageNet`

| Initial changing rate | Q=2000 | Q=4000 | Q=6000 | Q=8000 | Q=10000 |
|---|---|---|---|---|---|
| | Sparsity = 0.2% | | | | |
| $\lambda_0 = 0.03$ | 10.2% | 22.6% | 29.2 % | 35.6% | **41.4%** |
| $\lambda_0 = 0.05$ | **12%** | **23.6%** | **31.6%** | **36.6%** | 40.4% |
| | Sparsity = 0.4% | | | | |
| $\lambda_0 = 0.03$ | 31% | 53.6% | 65.6% | 74.2% | 80% |
| $\lambda_0 = 0.05$ | **33.2%** | **54.2%** | **66.8%** | **76%** | **82.4%** |
| | Sparsity = 0.6% | | | | |
| $\lambda_0 = 0.03$ | 45.4% | 75.4% | 84.6% | 89.8% | 92.8% |
| $\lambda_0 = 0.05$ | **45.4%** | **75.6%** | **87.4%** | **91.8%** | **94.6%** |

## M.3 THE CHOICE OF $\alpha^{prior}$

In this section, we discuss the choice of $\alpha^{prior}$ and provide an analysis on the convergence time.

- $\boldsymbol{\alpha^{prior}} = 1$ ($\alpha_i = 1$ where $i \in [1,k]$). In our proposal, we draw multiple pixels (equivalent to multiple elements in a binary matrix u) from the Categorical distribution ($K$ categories) parameterized by $\boldsymbol{\theta} = [\theta_1, \theta_2, ..., \theta_K]$. When initializing an attack, we have no prior knowledge of the influence of each pixel that is higher or lower than other pixels on the model's decision so it is sensible to assume all pixels have a similar influence. Consequently, all pixels should have the same chance to be selected for perturbation (to be manipulated). To this end, the Categorical distribution where multiple pixels are drawn from should be a uniform distribution and $\theta_1 = \theta_2 = ... = \theta_K = \frac{1}{K}$.

  We note that Dirichlet distribution is the conjugate prior distribution of the Categorical distribution. If the Categorical distribution is a uniform distribution, Dirichlet distribution is also a uniform distribution. In probability and statistics, Dirichlet distribution (parameterized by a concentration vector $\boldsymbol{\alpha} = [\alpha_1, \alpha_2 ..., \alpha_K]$, each $\alpha_i$ represents the i-th element where $K$ is the total number of elements) is equivalent to a uniform distribution over all of the elements when $\boldsymbol{\alpha} = [\alpha_1, \alpha_2 ..., \alpha_K] = [1, 1, ..., 1]$. In other words, there is no prior knowledge favoring one element over another. Therefore, we choose $\boldsymbol{\alpha^{prior}} = \mathbf{1}$.

- $\boldsymbol{\alpha^{prior}} < 1$ ($\alpha_i < 1$ where $i \in [1,k]$). We have $\boldsymbol{\alpha^{posterior}} = \boldsymbol{\alpha^{prior}} + s^{(t)}$ and $s^{(t)} = (a^{(t)}+z)/(n^{(t)}+z) - 1$. So we have $\boldsymbol{\alpha^{posterior}} = \boldsymbol{\alpha^{prior}} + (a^{(t)}+z)/(n^{(t)}+z) - 1$. Because $(a^{(t)} + z)/(n^{(t)} + z) \leq 1$, we cannot choose $\boldsymbol{\alpha^{prior}} < 1$ to ensure that the parameters controlling the Dirichlet distribution are always positive ($\boldsymbol{\alpha^{posterior}} > 0$).

- $\boldsymbol{\alpha^{prior}} > 1$ ($\alpha_i > 1$ where $i \in [1,k]$). Since $\boldsymbol{\alpha^{posterior}} = \boldsymbol{\alpha^{prior}} + (a^{(t)}+z)/(n^{(t)}+z) - 1$ and $0 < (a^{(t)}+z)/(n^{(t)}+z) \leq 1$, if $\boldsymbol{\alpha^{prior}} \gg 1$, in the first few iterations, $\boldsymbol{\alpha^{posterior}}$ almost remains unchanged so the algorithm will not converge. If $\boldsymbol{\alpha^{prior}} > 1$, the farther from 1 $\boldsymbol{\alpha^{prior}}$ is, the more subtle the $\boldsymbol{\alpha^{posterior}}$ changes. Now, the update $(a^{(t)}+z)/(n^{(t)}+z)$ needs more iterations (times) to significantly influence $\boldsymbol{\alpha^{posterior}}$. In other words, the proposed method requires more time to learn the Dirichlet distribution (update $\boldsymbol{\alpha^{posterior}}$). Thus, the convergence time will be longer. Consequently, the larger $\alpha_i$ is, the longer the convergence time is.

## N BRUSLEATTACK WITH DIFFERENT SCHEDULERS

We carry out a comprehensive experiment to examine the impact of different schedulers including cosine annealing and step decay. In this experiment, we use the same evaluation set with 1000 images from `CIFAR-10` evenly distributed in 10 classes and 9 target classes per image and we use the same query budget (500 queries). The results in Table 24 show the ASR at different sparsity levels. These results show that our proposed scheduler slightly outperforms all other schedulers. Based on the results, Step Decay or Cosine Annealing schedulers can be a good alternative.

**Table 24:** ASR comparison between using a Power Step Decay (our proposal) and other schedulers on `CIFAR-10`.

| Sparsity | Our Proposal | Step Decay | Cosine Annealing |
|---|---|---|---|
| 1.0% | **68.21**% | 68.11% | 68.02% |
| 2.0% | **90.24**% | 89.34% | 89.15% |
| 2.9% | **96.59**% | 96.12% | 95.89% |
| 3.9% | **98.48**% | 98.26% | 98.18% |

## O  ALGORITHM PSEUDOCODES

**Initialization.** Algorithm 2 presents pseudo-code for attack initialization as presented in Section 3.3.

---

**Algorithm 2:** Initialization

**Input:** source image $x$, synthetic color image $x'$ source label $y$, target label $y_{\text{target}}$
  number of initial samples $N$, perturbation budget $B$, victim model $f_M$

1   $\ell \leftarrow \infty$
2   **for** $i = 1$ **to** $N$ **do**
3     Generate $u'$ by uniformly enabling $B$ bits of $\mathbf{0}$ at random
4     $\ell' \leftarrow L(f_M(g(u'; x, x')), y^*)$
5     **if** $\ell' < \ell$ **then**
6       $u \leftarrow u'$, $\ell \leftarrow \ell'$
7   **end for**
8   **return** $u, \ell$

---

**Generation.** Algorithm 3 presents pseudo-code for generating new data point as presented in Section 3.3.

---

**Algorithm 3:** Generation

**Input:** probability $\theta$, bias map $M$, mask $u$, changing rate $\lambda$

1   $b \leftarrow \lceil (1 - \lambda)B \rceil$
2   $v_1 \ldots, v_b \sim \text{Cat}(v \mid \theta, u = \mathbf{1})$
3   $q_1 \ldots, v_{B-b} \sim \text{Cat}(q \mid \theta M, u = \mathbf{0})$
4   $u^{(t)} = [\vee_{k=1}^{b} v_k^{(t)}] \vee [\vee_{r=1}^{B-b} q_k^{(t)}]$
5   **return** $u$

---

**Update.** Algorithm 4 presents pseudo-code for updating an accepted mask (a solution in round $t$) and estimated $\theta^{(t)}$ as presented in Section 3.3 and illustrated in Figure 2.

## P  EVALUATION PROTOCOL

In this section, we present the evaluation protocol used in this paper.

1. In the targeted attack settings.
   - SparseRS (Croce et al., 2022) evaluation with ImageNet: Selected 500 source images. But each source image class was flipped to only one random target class using the attack. So that is a total of 500 source-image-to-target class attacks. This evaluation protocol may select the same target class to attack in the 500 attacks conducted. Thus, this could lead to potential biases in the results because some classes may be easier to attack than others.

---

**Algorithm 4:** Update

---

**Input:** previous mask and loss $\boldsymbol{u}^{(t-1)}, \ell^{(t-1)}$, current mask and loss $\boldsymbol{u}^{(t)}, \ell^{(t)}$, small constant $z$, matrices $\boldsymbol{a}^{(t)}, \boldsymbol{n}^{(t)}$

1  $\boldsymbol{a} \leftarrow \boldsymbol{a}^{(t)}, \boldsymbol{n} \leftarrow \boldsymbol{n}^{(t)}$
2  $n_{i,j \in \{[i,j] | (\boldsymbol{u}^{(t-1)} \vee \boldsymbol{u}^{(t)})_{i,j}=1\}}$ increase by 1
3  **if** $\ell^{(t)} < \ell^{(t-1)}$ **then**
4  $\quad \boldsymbol{u} \leftarrow \boldsymbol{u}^{(t)}, \ell \leftarrow \ell^{(t)}$
5  **else**
6  $\quad \boldsymbol{u} \leftarrow \boldsymbol{u}^{(t-1)}, \ell \leftarrow \ell^{(t-1)}$
7  $\quad a_{i,j \in \{[i,j] | (\boldsymbol{u}^{(t-1)} \oplus (\boldsymbol{u}^{(t-1)} \wedge \boldsymbol{u}^{(t)}))_{i,j}=1\}}$ increase by 1
8  **end if**
9  $s \leftarrow \frac{a+z}{n+z} - 1$
10 Update $\boldsymbol{\alpha}^{\text{posterior}}$ using $s$ and Equation 6
11 Update $\boldsymbol{\theta}$ using $\boldsymbol{\alpha}^{\text{posterior}}$ and Equation 8
12 **return** $\boldsymbol{u}, \ell, \boldsymbol{\theta}, \boldsymbol{a}, \boldsymbol{n}$

---

- To avoid the problem, in the targeted attack setting, we followed the evaluation protocol used in (Vo et al., 2022a). Essentially, we flip the label of the source image to several targeted classes, this can help address potential biases caused by relatively easier classes getting selected multiple times for a target class.
- Our evaluation with `ImageNet`: We randomly selected 200 correctly classified source images evenly distributed among 200 random classes. But, in contrast to SparseRS, we selected 5 random target classes to attack for each source image. In total we did $200 \times 5 = 1000$ source-image-to-target class attacks on `ImageNet` for targeted attacks.

2. In the untargeted attack setting (attacks against defended models), we conducted 500 attacks (similar to SparseRS). We randomly selected 500 correctly classified test images from 500 different classes for attacks.

3. Further, our unique and exhaustive testing with `CIFAR-10` and `STL-10` corroborates `ImageNet` results given the significant amount of resources it takes to attack the high-resolution `ImageNet` ($224 \times 224$) models.

   - For `STL-10` we conducted 60,093 attacks against each deep learning model (6,677 of all 10,000 images in the test set which are correctly classified versus 9 other classes as target classes for each source image). We used every single test set image in `STL-10` ($96 \times 96$) in our attacks to mount the exhaustive evaluation where no image from the test set was left out.
   - For `CIFAR-10` ($32 \times 32$) we conducted 9,000 attacks against each deep learning model (1000 random images correctly classified versus the 9 other classes as target classes for each source image).

4. For evaluations against a real-world system (GCV) in the significantly more difficult targeted setting (not the untargeted setting), we provide new benchmarks for attack demonstration because we provide a comparison between BRUSLEATTACK and the previous attack, SPARSE-RS. To make it clear, we provide a brief comparison as follows:

   - Other related past studies (dense attacks)(Ilyas et al., 2018; Guo et al., 2019), showcase an attack against a real-world system but uses 10 attacks. While Ilyas et al. (2018) illustrated only one successful example when carrying out an attack against Google Could Vision.
   - Importantly, we did not simply use our method only, as in (Ilyas et al., 2018; Guo et al., 2019) but demonstrated a comparison between BRUSLEATTACK and SPARSE-RS. In practice, we used 10 samples for each attack, so there are 20 attacks.

In general, our evaluation protocol is much stronger than the one used in previous studies. We evaluate on three different datasets `CIFAR-10`, `STL-10` (not evaluated in prior attacks) and `ImageNet` with ResNet-50, ResNet-50 (SIN), Visitation Transformer (not evaluated in prior attacks).

## Q    VISUALIZATIONS OF ATTACK AGAINST GOOGLE CLOUD VISION

Table 25 and Figure 11, Table 26 and Figure 12 show our attack against real-world system Google Cloud Vision API.

**Table 25:** Demonstration of sparse attacks against GCV in targeted settings. BRUSLEATTACK is able to successfully yield adversarial instances for all five examples with less queries than SPARSE-RS. Especially, for the example of Mushroom, SPARSE-RS fails to attack GCV within a budget of 5000 queries. Demonstration on GCV API (online platform) is shown in Figure 11.

| Examples | | | | | |
|---|---|---|---|---|---|
| No Attack | Car | Flower | Fire Truck | Vehicle | Mushroom |
| BRUSLEATTACK | Window (**1.8K** Queries) | Yellow Pepper (**99** Queries) | Window (**328** Queries) | Window (**1.83K** Queries) | Landscape (**490** Queries) |
| SPARSE-RS | Window (4.66K Queries) | Yellow Pepper (211 Queries) | Window (395 Queries) | Window (3.3K Queries) | Mushroom (>5K Queries) |

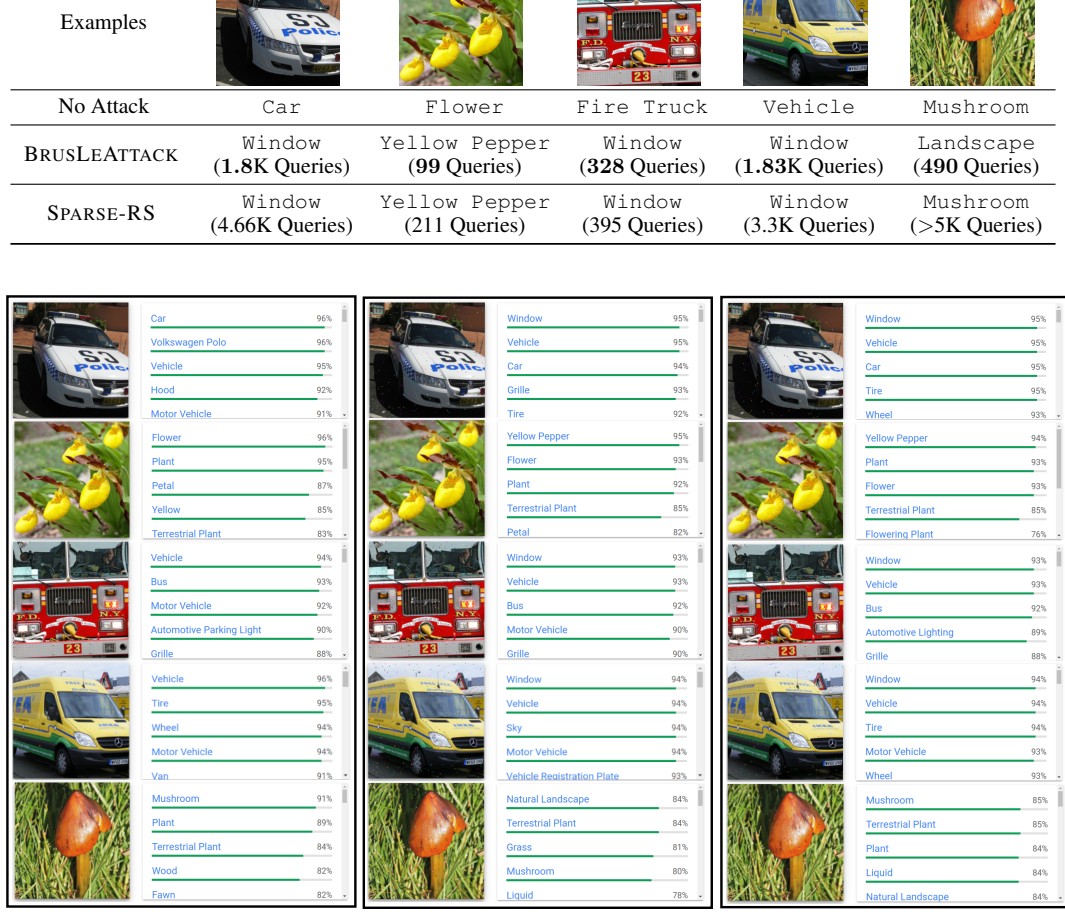

a)  No Attack (Clean input)    b)    Our Attack    c)    SparseRS

**Figure 11:** a) demonstrates results for clean image (no attack) predicted by Google Cloud Vision (GCV). b) shows the predictions from GCV for adversarial examples crafted successfully by BRUSLEATTACK with less than 3,000 queries and sparsity of 0.05 %. c) shows the results from GCV for adversarial examples crafted by SPARSE-RS with the same sparsity. But SPARSE-RS needs more queries than BRUSLEATTACK to successfully yield adversarial images or fail to attack with query budget up to 5,000 as shown in Table 25.

**Table 26:** Demonstration of sparse attacks against GCV in targeted settings. BRUSLEATTACK is able to successfully yield adversarial instances for all five examples with less queries than SPARSE-RS. Especially, for the example of `Mushroom`, SPARSE-RS fails to attack GCV within a budget of 5000 queries. Demonstration on GCV API (online platform) is shown in Figure 12.

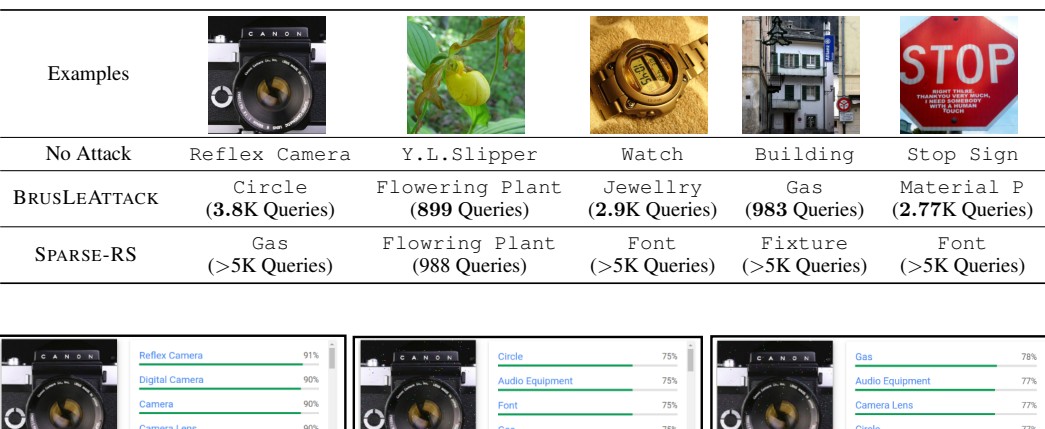

| Examples | | | | | |
|---|---|---|---|---|---|
| No Attack | Reflex Camera | Y.L.Slipper | Watch | Building | Stop Sign |
| BRUSLEATTACK | Circle (**3.8K** Queries) | Flowering Plant (**899** Queries) | Jewellry (**2.9K** Queries) | Gas (**983** Queries) | Material P (**2.77K** Queries) |
| SPARSE-RS | Gas (>5K Queries) | Flowring Plant (988 Queries) | Font (>5K Queries) | Fixture (>5K Queries) | Font (>5K Queries) |

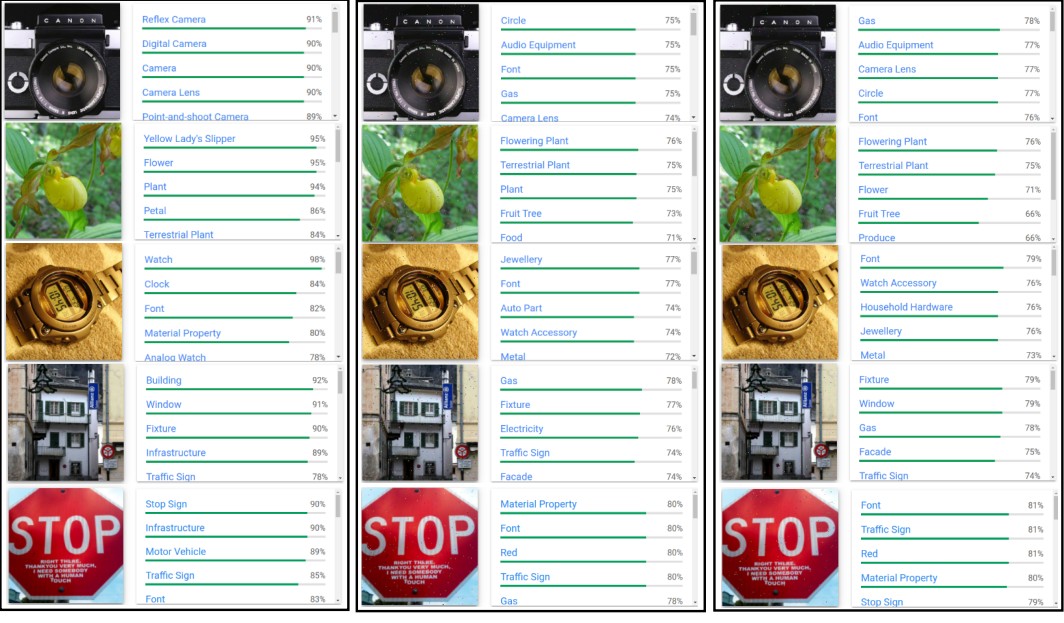

a)   No Attack (Clean input)    b)    Our Attack    c)    SparseRS

**Figure 12:** a) demonstrates results for clean image (no attack) predicted by Google Cloud Vision (GCV). b) shows the predictions from GCV for adversarial examples crafted successfully by BRUSLEATTACK with less than 3,000 queries and sparsity of 0.05 %. c) shows the results from GCV for adversarial examples crafted by SPARSE-RS with the same sparsity. But SPARSE-RS need more queries than BRUSLEATTACK to successfully yield adversarial images or fail to attack with query budget up to 5,000 as shown in Table 25.

# R    VISUALIZATIONS OF SPARSE ADVERSARIAL EXAMPLES AND DISSIMILARITY MAPS

In this section, we illustrate:

- Sparse adversarial examples, sparse perturbation crafted by our methods versus salient region produced by GradCAM method Selvaraju et al. (2017) or attention map produced by a ViT model Dosovitskiy et al. (2021).
- Sparse adversarial examples crafted by BRUSLEATTACK when attacking ResNet-50, ResNet-50 (SIN) and Vistion Transformer.
- Dissimilarity Map produced from a pair of a source and a synthetic color images.

Figure 13 and 14 illustrate sparse adversarial examples and spare perturbation of images from `ImageNet` in targeted and untargeted settings. In targeted settings, we use a query budget of 10K, while in untargeted settings, we set a query limit of 5K. We use GradCAM and Attention Map from ViT to demonstrate salient and attention regions. The sparse perturbation $\delta$ is the absolute difference between source images and their sparse adversarial. Formally, sparse perturbations can be defined as $\delta = |\boldsymbol{x} - \tilde{\boldsymbol{x}}|$.

The results show that for ResNet-50, the solutions found do not need to perturb salient regions on an image to mislead the models (both targeted and untargeted attacks). Attacks with ViT models in untargeted settings also lead to a similar observation. Interestingly, for some images e.g. a snake or a goldfinch in Figure 13, we observe that a set of perturbed pixels yielded by our method is more concentrated in the attention region of ViT. This seems to indicate some adversarial solutions achieves their objective by degrading the performance of a ViT. This is perhaps not an unexpected observation, given the importance of attention mechanisms to transformer models.

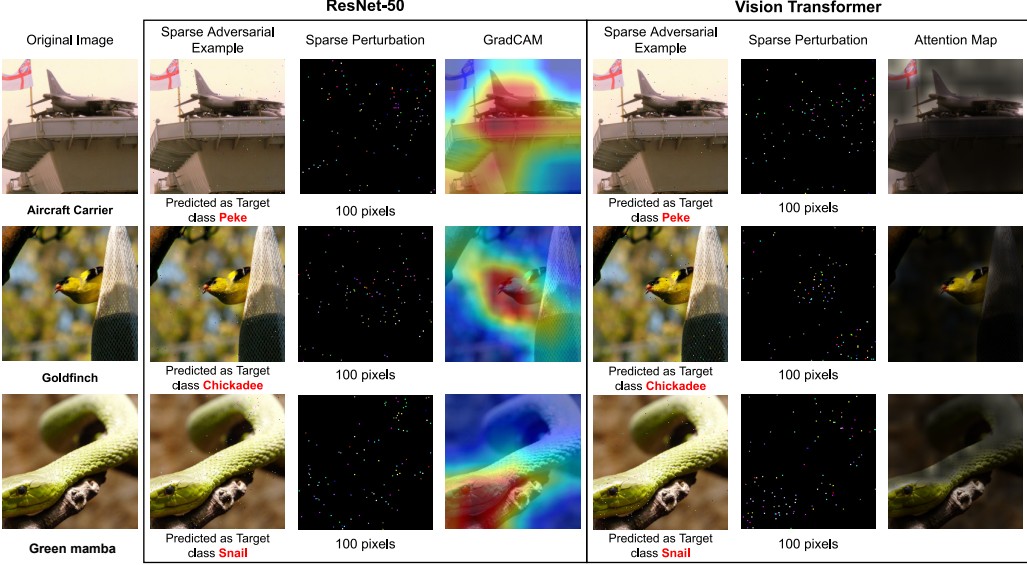

**Figure 13: Targeted Attack.** Visualization of Adversarial examples crafted by BRUSLEATTACK with a budget of 10K queries. In the image of sparse perturbation, **each pixel is zoomed in 9 times ($9\times$)** to make them more visible.

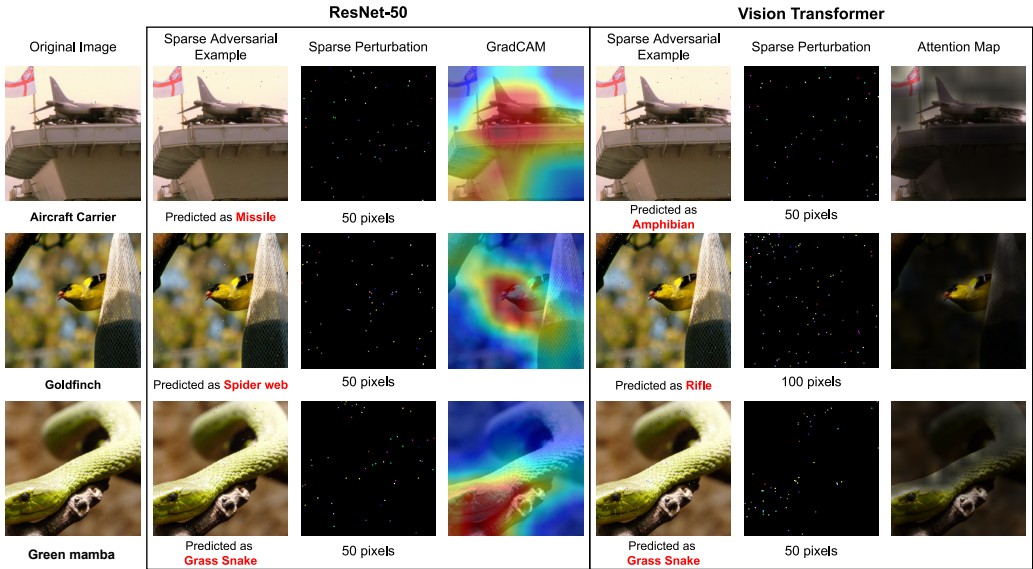

**Figure 14: Untargeted Attack.** Visualization of Adversarial examples crafted by BRUSLEATTACK with a budget of 5K queries. In the image of sparse perturbation, **each pixel is zoomed in 9 times** ($9\times$) to make them more visible.

Figure 15 and 16 demonstrate some examples of adversarial examples yielded by BRUSLEATTACK when attacking different deep learning models (ResNet-50, ResNet-50 (SIN) and Vision Transformer) in targeted settings produced using a 10K query budget.

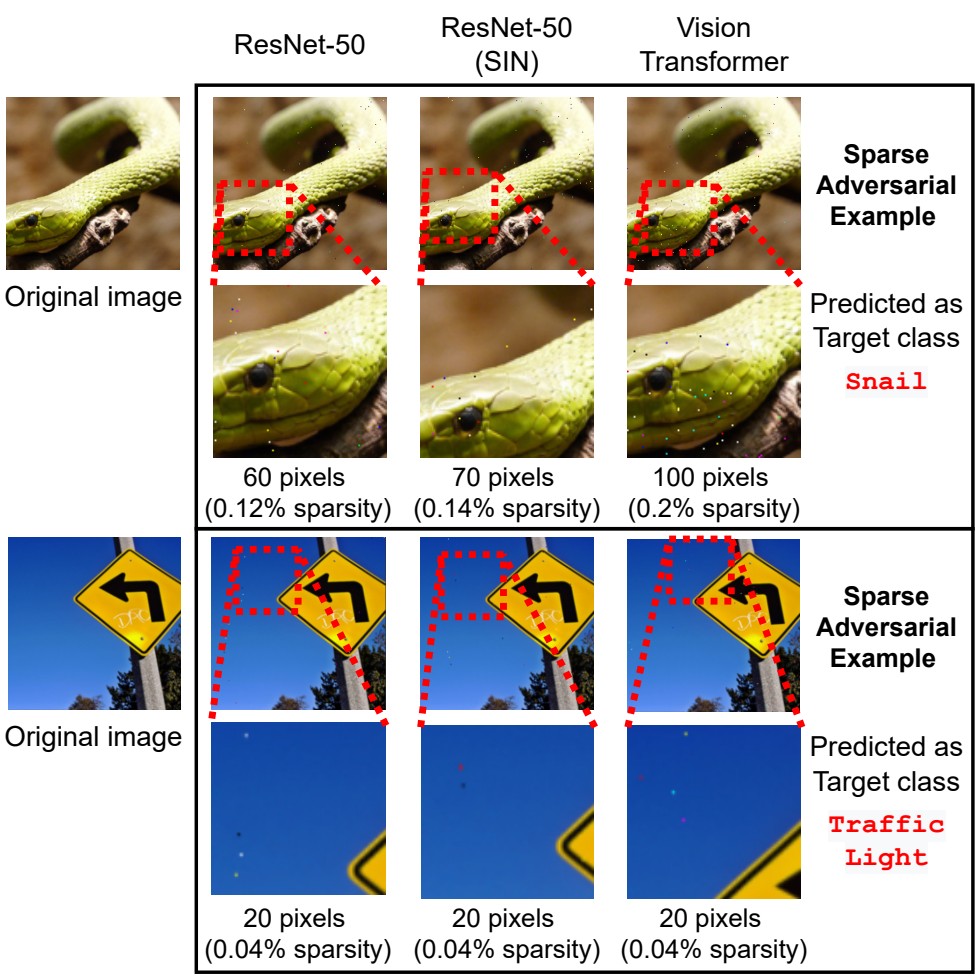

**Figure 15:** Visualization of Adversarial examples crafted by BRUSLEATTACK with a budget of 10K queries.

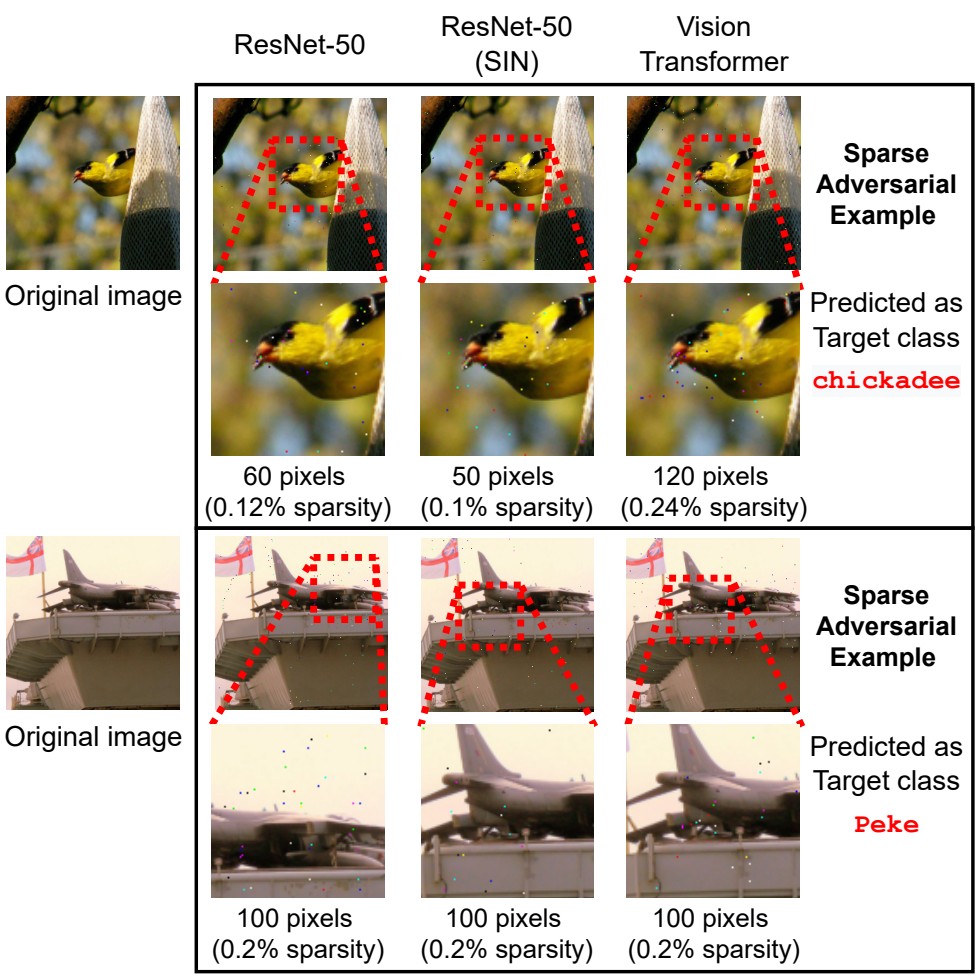

**Figure 16:** Visualization of Adversarial examples crafted by BRUSLEATTACK with a budget of 10K queries.

Figure 17 illustrates some examples of Dissimilarity Map yielded by a source image and a synthetic color image.

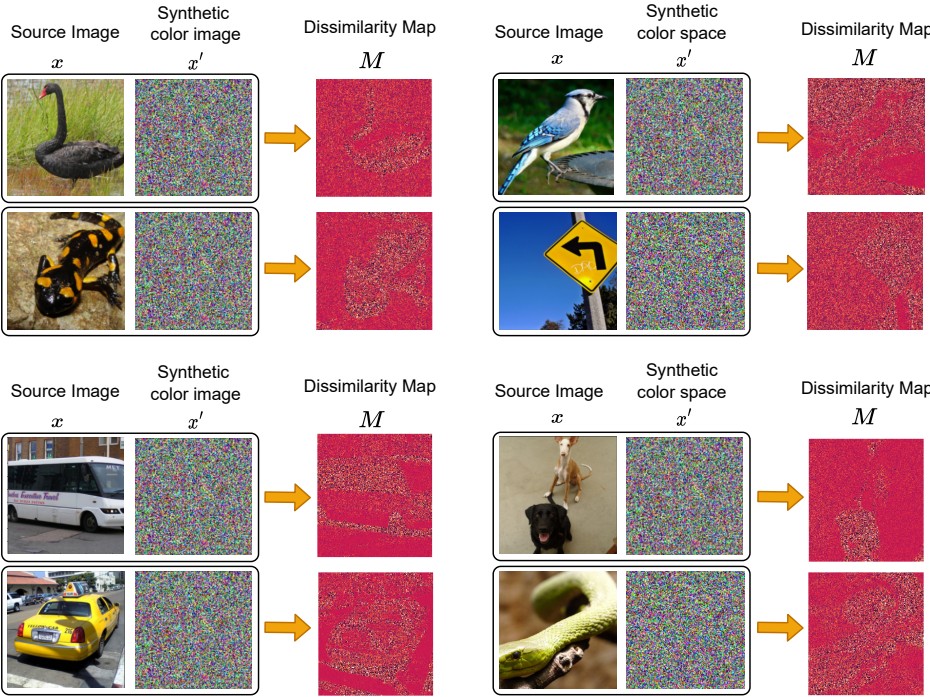

**Figure 17:** Visualization of Dissimilarity Maps between a source image and a synthetic color image.

