# OpenReview forum: "BRUSLEATTACK: A QUERY-EFFICIENT SCORE- BASED BLACK-BOX SPARSE ADVERSARIAL ATTACK"
_ICLR.cc/2024/Conference — ICLR 2024 poster_

### Official Review · Reviewer_UEvH · 2023-10-31

**Soundness:** 3 good
**Presentation:** 2 fair
**Contribution:** 3 good
**Rating:** 6
**Confidence:** 4

**Summary:**

This paper studies score-based sparse adversarial attacks against deep learning models. The problem is challenging due to the combinatorial optimization problem and black-box setting. The paper introduces a novel attack algorithm by reducing the search space of adversarial attacks and proposes a probabilistic method to optimize the binary matrix. The extensive experiments demonstrate the query efficiency and effectiveness of attacking black-box models, including black-box APIs.

**Strengths:**

- The problem formulation is interesting and the method is solid.
- The experiments demonstrate the effectiveness of the proposed method.

**Weaknesses:**

- Some technical details are unclear. For example, the authors have stated that they reduce the search space by introducing a synthetic color image. However, it is not clear how this approach can reduce the search space. Although the authors have provided detailed analyses in Appendix, they should provide some intuitions in the main paper to help understand the paper.
- Learning a distribution of the optimization parameters seems to be a natural way. Natural evolution strategy is commonly adopted for score-based attacks by learning a search distribution, although the previous work adopts Gaussian distribution for $\ell_2$ and $\ell_\infty$ norms. The authors are encourages to discuss with NES. Also, a recent work ("Black-box Detection of Backdoor Attacks with Limited Information and Data", ICCV 2021) also proposes a score-based method to optimize sparse perturbations, although it studies a different problem. More discussions and comparisons are needed.

**Questions:**

- Can this method be extended to adversarial patch attack, which is a special case of $\ell_0$ perturbations but is physically realizable.
- Could you show some sparse adversarial examples? Are there any interesting observations on what pixels are most influential for changing the prediction?

---

> ### Author Response · Authors · 2023-11-20
> **Response to Reviewer UEvH**
>
> Q1.  It is not clear how this approach can reduce the search space?
>
> We are sorry if the details in __Section 3.1__ are not clear enough in the main paper (detailed explanations are in __Appendix G and H__), we will improve the explanation in the main paper for the camera ready. Till then:
>
> *How does the synthetic color image reduce the search space?*
>
> - Consider, each pixel has c channels (c = 3 in RGB images). If each color scale has $2^m$ (e.g. m = 8) values, the color of each channel of each pixel can be one of $2^m$. If  image size is $w\times h\times c$, the total possible values (a combination of all pixels and all color values) is $(2^m)^{w\times h\times c}$.
>
> - Now, to search for a solution that is a set of ‘n’ perturbed color pixels in a source image, we would need to search for ‘$n$’ pixels and their colors. So, we need to select $c\times n$ values from $2^{m\times w\times h\times c}$. Therefore, the search space is equivalent to $C^{c\times n}_{2^{m\times w\times h\times c}}$.
>
> - In our method, we only search for ‘$n$’ pixels. For the c color values,  we select them from their corresponding pixel positions in the synthetic color image ($w\times h\times c$). The synthetic image we generate before the attack is not altered during the attack. In this way, all c channels of these pixels are effectively altered together. So, we do not need to search color values but only search for n pixels (select $n$ values from $w\times h$). Therefore, the search space in our method is equivalent to $C^n_{w\times h}$. This is a significant reduction e.g. from search space $C^{c\times n}_{2^{m\times w\times h\times c}} $
>
> to $C^n_{w\times h}$.

---

> ### Author Response · Authors · 2023-11-20
> **Response to Reviewer UEvH**
>
> Q2. the authors are encourages to discuss with NES. a recent work ("Black-box Detection of Backdoor Attacks with Limited Information and Data", ICCV 2021) also proposes a score-based method to optimize sparse perturbations, although it studies a different problem?
>
> Thank you for your suggestion.
>
> Natural Evolution Strategies (NES) is a family of black-box optimization methods that learns a search distribution by employing an estimated gradient on its distribution parameters [6, 7]. As you mentioned, NES was adopted for score-based dense (L2 and L∞ norms) attacks in (ICML 2018) [8] since they mainly adopted a Gaussian distribution for continuous variables. However, solving the problem posed in sparse attacks involving both discrete and continuous variables. And, leads to an NP-hard problem [9, 10]. Therefore, naively adopting NES for sparse attacks is non-trivial.
>
> __Comparison (NES, ICCV’21)__
>
> The work B3D in (ICCV 2021) [7], in a defense for a data poisoning attack or backdoor attack, proposed an algorithm to reverse-engineer the potential Trojan trigger used to activate the backdoor injected into a model. Although the method is motivated by NES and operates in a score-based setting involving both continuous and discrete variables, as with a sparse attack problem they are designed for completely different threat models (backdoor attacks with data poisoning versus adversarial attacks). So, it is hard to compare both of them properly. Further there is __*no public code release*__ to attempt to adapt this algorithm for a sparse attack.
> Adopting the score-based components in [7] for adversarial attacks is interesting. In fact, we also found adopting our method to reverse-engineer a backdoor trigger interesting. Therefore, both of these avenues are interesting research directions that we can pursue in the future. We will clarify this in our camera ready version.
>
> However, more qualitatively, there are a number of differences between our approach and those elements in [7].
>
> - __Method and Distribution differences__: [7] learns a search distribution determined by its parameters through estimating the gradient on the parameters of this search distribution. In the meantime, our approach is to learn a search distribution through Bayesian learning. While [7] employed Bernoulli distribution for working with discrete variables, we used Categorical distribution to search discrete variables.
> - __Search space (larger vs. smaller)__: B3D searches for a potential Torjan trigger in an enormous space as it requires to search for pixels’ position and color. Our approach reduces the search space and only searches for pixels (pixels’ position) to be altered so our search space is significantly lower than the search space used in [7] if the trigger size is the same as the number of perturbed pixels.
> - __Perturbation pattern (square shape vs. any set of pixel distribution)__: [7] aims to search for a trigger that usually has a size of 1x1, 2x2 or 3x3 so the trigger shape is a small square. In contrast, our attack aims to search for a set of pixels that could be anywhere in an image and the number of pixels could be varied tremendously (determined by desired sparsity). Thus, the combinatorial solutions in a sparse attack problem can be larger than the one in [7] (even when we equate the trigger size to the number of perturbed pixels).
> - __Query efficiency (is a primary objective vs. not an objective)__: Our approach aims to search for a solution in a query-efficiency manner while it is not clear how efficient the method is to reverse-engineer a trigger.
>
> We will clarify all of this in our camera ready.
>
> Q3. extended to adversarial patch attack?
>
> Thank you for the interesting idea. We showed that our method can attack online models such as Google Cloud vision but extending our method to generate a printable adversarial patch whilst minimizing the number of queries will indeed lead us towards an interesting and promising research direction in the future.
>
> Q4. Could you show some sparse adversarial examples?
>
> - Please see __*Figure 13*__ and __*Figure 14*__ in our Appendix (__*newly added*__)
> - Please see __*Figure 15*__ and __*Figures 16*__ (previously numbered as 13 and 14).
>
> *References*
>
> [6] D Wierstra, T Schaul, T Glasmachers, Y Sun, J Schmidhuber, Natural Evolution Strategies, Journal of Machine Learning Research 2014.
>
> [7] Y Dong, X Yang, Z Deng, T Pang, Z Xiao, H Su, J Zhu, Black-box Detection of Backdoor Attacks with Limited Information and Data, ICCV, 2021,
>
> [8] A. Ilyas, L. Engstrom, A. Athalye, and J. Lin. Black-box adversarial attacks with limited queries 351 and information. ICML, 2018.
>
> [9] A Modas, S-M Moosavi-Dezfooli, and P Frossard. Sparsefool: a few pixels make a big difference. CVPR, 2019.
>
> [10] X. Dong, D. Chen, J. Bao, C. Qin, L. Yuan, W. Zhang, N. Yu, and D. Chen. GreedyFool: DistortionAware Sparse Adversarial Attack. NeurIPS, 2020.

---

> > ### Comment · Reviewer_UEvH · 2023-11-22
> > **Thanks for the response**
> >
> > Thank you for providing the responses, which have addressed most of my concerns. The authors are encouraged to further revise the paper by adding the discussions and results.

---

> ### Author Response · Authors · 2023-11-23
> **We have uploaded a new PDF**
>
> We are happy to report that we've updated our paper.
> - Please see blue highlights in the Appendices that have been re-organized with *new results*, *discussions* and *visualizations* added in.  Thank you for your valuable input, we really appreciate it.
> - Especially, see the __*new visualization*__ we added in response to Q4 (Fig. 13 and 14 in __*Appendix R*__).
> - We will update a new version with presentation improvements to the main text, next.

---

> ### Public Comment · ~Damith_Ranasinghe1 · 2024-05-09
> **Meta Review Comments Addressed (Suggested Ablation)**
>
> We have addressed the suggested ablation of Sparse-RS in the final version of their paper. Please see the new study in the Appendix.
>
> * ***Appendix E.2*** Demonstrating the impact of the Bayesian framework based search (Comparison with an adapted SPARSE-RS using our synthetic images.
>
> Thank you.
> The Authors.

---

### Official Review · Reviewer_tLev · 2023-11-05

**Soundness:** 2 fair
**Presentation:** 2 fair
**Contribution:** 3 good
**Rating:** 5
**Confidence:** 4

**Summary:**

This paper introduces a black-box attack method to generate sparse adversarial perturbations to fool the model. The method focuses on learning the masks representing the perturbed pixels and use Bayesian optimization to achieve it. Extensive experiments and comparison with the baseline demonstrate the effectiveness of the algorithm.

**Strengths:**

The motivation is well-justified. The observation that interpolation with a real image and a synthetic image can generate sparse adversarial examples is interesting. The Bayesian optimization in this context is novel. The experiments comprehensively demonstrate the effectiveness of the algorithm.

**Weaknesses:**

Despite the strengths above, the paper has the following weaknesses:

1. [Presentation] The notation of the paper is a bit confusing. I suggest the authors put a notation table in the appendix to facilitate the reader to better understand every notation in the paper.

2. [Theorem] I did not understand why proof in Appendix G matters. The re-parameterization is straightforward.

3. [Experiments] Regarding the efficiency of the algorithm, using the wall clock time may be a better choice than the number of queries. This is because the proposed algorithm may have a higher complexity per query.

4. [Code] No sample code is provided, there is a reproduction concern.

**Questions:**

In addition to the weakness part above, I have the following additional questions:

1. In addition to the $l_2$ and $l_\infty$ robust models, I suggest the authors try evaluating the attack perturbation for $l_1$ robust models, since $l_1$ norm ball the convex hull of the perturbations bounded by $l_0$ norms. Possible baselines include strong AA-$l_1$ [A] and efficient Fast-EG-$l_1$ [B].

[A] Croce, Francesco, and Matthias Hein. "Mind the box: $ l_1 $-APGD for sparse adversarial attacks on image classifiers." International Conference on Machine Learning. PMLR, 2021.

[B] Jiang, Yulun, et al. "Towards Stable and Efficient Adversarial Training against $ l_1 $ Bounded Adversarial Attacks." International Conference on Machine Learning. PMLR, 2023.

The current manuscript is a borderline paper, I will conduct another round of evaluation after the discussion with the authors.

---

> ### Author Response · Authors · 2023-11-20
> **Response to Reviewer tLev**
>
> Thank you for your helpful feedback.
>
> 1. __[Presentation]__: We will put a notation table in our camera-ready version.
> 2. __[Theorem]__: We agree the reparameterization is straightforward. We added the proof in Appendix G for completeness and it can be helpful to make our new problem formulation more understandable. We feel some readers may still request this.
> 3. __[Experiments]__: We used the number of queries because.
>     - First, in practice, the lower the number of queries is, the lower the cost of an attack is. For instance, Google cloud vision API (label, logo, facial or landmark detection) costs \\$1.5 per 1K queries (https://cloud.google.com/vision/pricing). If an attack requires 10K, it costs \\$15 per attack sample. In our experiments, we evaluated 1000 attacks with a budget of 10K queries/attack. The total cost for this is \\$15K.
>     - Second, providers can recognize a large number of queries with similar inputs made in rapid succession to detect malicious activity and thwart query attacks. If an attack requires more queries, it is more likely to be detected and be thwarted.
>     - Third, assessing queries is the measure of algorithm efficiency largely adopted by black-box adversarial attack researchers.
>
> But, we followed your suggestion and measured the wall clock time per query between BruSLeAttack (ours) and Sparse-RS to show the efficiency of the algorithm in terms of attack time.
> - We tested both methods with the same attack input for the ImageNet task using the ResNet-50 model in the targeted attack setting. We ran and measured the total time for 10,000 queries. We repeated this experiment 10 times and computed the averages below:
>     - BruSLeAttack: The average is 14.45 ms/query.
>     - SparseRS: The average is 17.89 ms/query.
> - As we can see, our method is actually slightly faster than the existing method.
> 5. __[Code]__: We will do a full open-source release at `https://brusliattack.github.io/` after the rebuttal and before the camera-ready version.
>
>
>
> 6. Evaluation against $l_1$ robust model?
>
>     - We have now evaluated our attack method’s robustness against $l_1$ robust models (Please note in the limited time we have only been able to run a few experiments, but we will update more results when that becomes available).
>     - Although both [A] and [B] illustrated their robustness against $l_1$ attacks, Fast-EG-1 is the current state-of-the-art method (as shown in [B] published in ICML2023). Therefore, in the limited time, we selected the $l_1$ robust model trained by the Fast-EG-1 method for our experiment.
>     - Similar to other experiments on CIFAR-10 against adversarial trained models ($l_2$ and $l_\infty$), we used 1000 correctly classified samples evenly distributed across 10 different classes.
>     -We *added* the results to __*the existing Table 8*__ in Appendix E and *reproduced* it here for completeness as Table 2. The results show that our attack *outperforms* Sparse-RS across different sparsity levels. With this addition, our evaluation against $l_1, l_2$ and $l_\infty$ robust models shows that our attack achieves better performance than Sparse-RS.
>     - We will add the new results and discussions to our Camera-Ready version.
>
> Table 2: A robustness comparison (lower ↓ is stronger) between SPARSE-RS and BRUSLEATTACK against $l_1, l_2$ and $l_\infty$ robust models on CIFAR-10. The attack robustness is measured by the degraded accuracy of models under attacks at different sparsity levels.
>
> |          | Undefended | Model   | $l\_\infty$ Robust |Model    | $l\_2$ Robust|Model | *(New Result)*  $l\_1$ Robust |Model |
> |---|---|---|---|---|---|---|-----|---|
> |Sparsity|Sparse-RS|**BruSLeAttack**|Sparse-RS|**BruSLeAttack**|Sparse-RS|**BruSLeAttack**|Sparse-RS|**BruSLeAttack**|
> |  0.39% | 26.5% | **24.2%** | 65.9% | **65.0%** |84.7% |**84.2%** |86.6% |**85.8%** |
> |  0.78% | 7.8% | **6.4%** | 48.1% | **46.0%** |70.6% |**68.3%** |75.8% |**74.8%** |
> |  1.17%  | 2.5% | **2.0%** | 38.1%  | **35.1%** |57.6% |**54.3%** |68.5% |**64.8%** |
> |  1.56%  | 0.6% | **0.6%** | 28.8% |**26.4%** |44.4% |**43.8%** |59.4% |**55.9%** |

---

### Official Review · Reviewer_kfkC · 2023-11-05

**Soundness:** 2 fair
**Presentation:** 1 poor
**Contribution:** 3 good
**Rating:** 6
**Confidence:** 3

**Summary:**

The authors first highlight the problems related to generating strong sparse l0 norm attacks. The non differential search space makes the problem NP hard. To generate a stronger attack, the authors propose an iterative algorithm which aims to estimate a mask made up of zeros and ones to determine whether to take the pixel from the original image or from a synthetically generated random image. The authors provide an analysis on different sampling distributions considered to generate synthetic images. Once the synthetic image is generated, the authors propose to sample each value of the mask from a categorical distribution parameterized by a training parameter \theta and conditioned on the difference between the original image and the synthetic image. Then the authors propose to estimate \theta by initializing it as a Dirichlet prior parameterized by \alpha_{i} for i \in {0,k} and updating it by taking an expectation over the posterior distribution of \theta conditioned on \alpha_{i} and the mask of last time step. Continuing this iteratively, the authors approximate the mask with zeros and ones within the threat l0 threat model. The authors demonstrate that the proposed approach is faster when compared to [1], scales to larger datasets like Imagenet and achieves improved performance over existing methods.

[1] Croce, Francesco et al. “Sparse-RS: a versatile framework for query-efficient sparse black-box adversarial attacks.” AAAI Conference on Artificial Intelligence (2020).

**Strengths:**

* The proposed algorithm seems interesting and innovative.
* The results demonstrate significant improvements over prior works.
* The proposed method shows improved performance on standard datasets like CIFAR10 and scales to larger scale datasets like Imagenet.

**Weaknesses:**

* I think the presentation of the paper needs improvement. For instance, in the introduction, it seems difficult to understand what authors are trying to say in “Our Proposed Algorithm” part. The authors should try to built a flow in the writing which can help them in explaining their method. Similarly the authors should not use long captions below the figures. This makes it difficult to follow what the authors are trying to convey. The functions GENERATION and INITIALIZATION in algorithm-1 should be defined properly.

* Since the authors only aim to generate the mask with 0’s and 1’s and thus replace the pixels values corresponding to mask of 1 from a synthetically generated random image, it is likely that the strength of the attack could be enhanced by adjusting the pixel values. Thus the proposed attack though performs better than existing methods, doesn't seem like an optimal attack and shows scope for improvement. It would be nice if the authors could share the effect of first running their proposed algorithm to find the pixels to be replaced and later changing the values of the pixels to make the attack stronger. Is it possible to enhance the strength of the attack by using this second step?

* I think some of the design choices in the proposed algorithm need more discussion. For instance, the algorithm might be sensitive to the initial values of Dirichlet parameters \alpha_{k}. Could the authors present some analysis on how the convergence time and attack success rate of the proposed method varies with the initial values of \alpha_{k}. Further, does the algorithm converge to same set of values of \alpha_{k} for different initial values?

Post Rebuttal: I would like to thank the authors for providing a comprehensive rebuttal. The authors have tried to improve the clarity of this work and also addressed my concerns. Therefore, I increase my score.

**Questions:**

It is not clear how the authors ensure that the attack follows the threat model. Could the authors clarify this. It would be nice if the authors can answer the concerns and questions raised in the weakness section

---

> ### Author Response · Authors · 2023-11-20
> **Response to Reviewer kfkC**
>
> Q1. In the introduction, it seems difficult to understand and the functions GENERATION and INITIALIZATION in algorithm-1 should be defined properly?
>
> - Thank you for your helpful feedback to improve our presentation.
>
> - We will revise the _“Our Proposal Algorithm”_ part and shorten some figures’ captions to make our method and paper easier to read.
> - We are sorry we missed highlighting that the pseudocode for the algorithms are in  __Appendix O__. We will revise the content in __Section 3.1__ to improve clarity and clearly highlight that pseudocode is located in __Appendix O__.
>
> Q2. Is it possible to enhance the strength of the attack by using this second step?
>
> - Indeed, an optimal search should also search for the optimal perturbation (value of pixels). Unfortunately, as we mention in Section 1 even without searching for the optimal perturbation, the problem is already NP-hard (see discussion in __Appendix G and H__). So, the real challenge is to find avenues to mitigate the complexity posed by the NP-hard problem.
>
> - The suggestion from the reviewer is interesting. Unfortunately, it won’t improve an initial solution, e.g. a set of pixel positions (meeting the l0 constraint). Because, that solution would already be an adversarial example as the optimisation problem would need this objective to guide the search. So, in searching for the position, you are effectively only considering solutions that are always adversarial. Subsequently, changing the values of pixel colors at those positions will not make it more adversarial, as the solution found is already able to fool the model to the target class.
>
> - In general, *all* our attempts to change the color of the pixels in searches led to significant increases in query budgets. In contrast, our approach aims to *model the influence of each pixel* bearing *a specific color*, probabilistically, and learn the probability model through the historical information collected from pixel manipulations. So, we chose not to search for pixels’ position and then, search for their color but we aim to do both simultaneously. In other words, the solution found by our method is a set of pixels with their specific colors.
>
> We hope the discussion makes things clear and we will add this extra clarification to the camera-ready version.

---

> ### Author Response · Authors · 2023-11-20
> **Response to Reviewer kfkC**
>
> Q3. Some of the design choices in the proposed algorithm need more discussion?
>
> We agree and thank you for these valuable suggestions. We included these in the Appendices.
> - __Appendix K__ (Dissimilarity Map)
> - __Appendix M__ (Hyper-parameters)
> - __Appendix N__ (Different Schedulers).
> - __Appendix M.3__. Discusses the initial values of parameter alpha ($\alpha^{prior}$) and the choice of this parameter ($\alpha^{prior}$=1). We will summarize it here and provide additional analysis if $\alpha^{prior}$ > 1 and $\alpha^{prior}$<1.
>
> __Appendix M.3__ (A summary of why we chose $\alpha^{prior}$=1?)
> - Initialization of the Dirichlet distribution: Since we have no prior knowledge of the influence of each pixel on the model’s decision when initialization, it is sensible to assume the influence of all pixels is similar. Thus, all pixels have the same chance to be selected and the Categorical distribution should be a uniform distribution ($\theta_1 = \theta_2 = … = \theta_K = \frac{1}{K}$). Since Dirichlet distribution is the conjugate prior distribution of the Categorical distribution, if the Categorical distribution is a uniform distribution, Dirichlet distribution is also a uniform distribution.
> - Now, in probability and statistics, Dirichlet distribution (parameterized by a concentration vector $\alpha =[\alpha_1, \alpha_2 … \alpha_K]$, where each $\alpha_i$ represents the i-th element where K is the total number of elements) is equivalent to a uniform distribution over all of the elements when $\alpha =[\alpha_1, \alpha_2 … \alpha_K]= [1, 1, …, 1]$. In other words, there is no prior knowledge favoring one element over another.
> - Therefore, we choose $\alpha =[\alpha_1, \alpha_2 … \alpha_K]= [1, 1, …, 1]$ and that is $\alpha^{prior}$.
>
> Why we cannot set $\alpha^{prior}$ < 1 ($\alpha_i<1$ where $i \in [1,k]$):
> - We have $\alpha^{posterior}  = \alpha^{prior}  + s^{(t)}$ and $s^{(t)} = (a^{(t)} + z)/(n^{(t)}+z) - 1$
> - So we have $\alpha^{posterior}  = \alpha^{prior} + (a^{(t)}+ z)/(n^{(t)}+z) - 1$. Because $(a^{(t)} + z)/(n^{(t)}+z)≤1$, we cannot choose $\alpha^{prior}$ < 1 to ensure that the parameters controlling the Dirichlet distribution are always positive ($\alpha^{posterior}$ > 0).
>
> What we do not set $\alpha^{prior}$ > 1 ($\alpha_i>1$ where $i \in [1,k]$):
> - Since $\alpha^{posterior}$  = $\alpha^{prior}+ (a^{(t)} + z)/(n^{(t)}+z) - 1$ and $0<(a^{(t)} + z)/(n^{(t)}+z)≤1$, if $\alpha^{prior}$ >> 1, in first few iteration, $\alpha^{posterior}$  almost remains unchanged so the algorithm will not converge. If $\alpha^{prior}$  > 1, the farther from 1 $\alpha^{prior}$ is, the more subtle the $\alpha^{posterior}$  changes. Now, the update $(a^{(t)} + z)/(n^{(t)}+z)$ needs more iterations (times) to significantly influence $\alpha^{posterior}$. In other words, the proposed method requires more time to learn the Dirichlet distribution (update $\alpha^{posterior}$). Thus, the convergence time will be longer. Consequently, the larger $\alpha_i$ is, the longer the convergence time is.
>
> We hope our analysis above makes things clear and we will add this extra analysis to the camera-ready version.

---

> ### Author Response · Authors · 2023-11-20
> **Response to Reviewer kfkC**
>
> Q4. How the authors ensure that the attack follows the threat model. Could the authors clarify this?
>
> In this paper, we focus on sparse attacks in score based settings (threat model) in which attackers have only access to the output information (confidence scores) and exploit only this information to search for a sparse adversarial example.
>
> To ensure that our attack follows the threat model, we summarize the process of our attack as follows:
>
> - An adversary has a desirable *source image* for which they would like to construct an adversarial example for. They can send the image as an input to the model and obtain the scores generated by the model. The adversary does not possess any other image or information.
> - Our algorithm code alters some pixels of the *source image* by replacing some pixels with their corresponding pixels from a synthetic color image to yield an altered image (see __Algorithm 3 (Generation)__).
> - Then, we submit this image as an input to the target model (a black-box classifier) and obtain *only* the confidence score vector and return this to our attack algorithm.
> - The confidence scores returned from the model are used to calculate a loss (__Equation 2__). *Please note that this loss is not the loss used for training a model*.
> - Then, our algorithm uses the information from the loss computation (increase or decrease in the loss in Equation 2) to learn the influence of pixels (see __Algorithm 4 (Update)__).
> - Subsequently, our algorithm uses the newly learned information together with what is known before to generate a new solution or adversarial example to test by sending the image to the black-box classifier.
>
> In general, our attack does not require *any prior knowledge of the victim model* such as model weight and parameters or model architecture. The process of our attack shown in __Figure 3__ and described in __Algorithm 1__ demonstrates that our attack clearly follows our threat model. We hope our clarification above clarifies things and will add these clarifications to the camera-ready version.
>
> Importantly we show that our attack __*can operate under our threat model*__ by attacking the online model from __Google Cloud Vision__. We wrote code to submit queries and obtain confidence scores from this service together with our algorithm to generate the results in __Figure 6__.

---

### Official Review · Reviewer_Xni7 · 2023-11-06

**Soundness:** 2 fair
**Presentation:** 3 good
**Contribution:** 2 fair
**Rating:** 3
**Confidence:** 4

**Summary:**

The paper proposes a new score-based blackbox adversarial attack under the sparse constraint (l_0 norm). Specifically, it first initial a random image to get the perturbation strength on every pixel. It then mix the random image with the original image using a learnable mask. By limiting the number of non-zero element in the mask under the l_0 norm constraint, it uses bayesian optimization to find the optimal position to generate an adversarial example on the sparsity constraint. The experiments show the proposed method could achieve a better attack success rate under different models, datasets and defensive models.

**Strengths:**

1. The paper is well-written and easy to follow.
2. The paper provides some interesting findings regarding sparse adversarial attack.

**Weaknesses:**

1. The proposed optimization is only conducted on finding the optimal position given a random initialized image x'. In other words, the proposed method is heavily based on the quality of the initialized x' and it heavily affects the optimality condition of the proposed method. I believe a successful sparse attack should not only find the best position but also find the optimal perturbation for those spaces.
2. The comparison with PGD_0 is not fair to me. The PGD_0 needs to find the minimum number of pixels to be perturbed where the proposed method only finds the best solution under certain threshold. I believe a fair comparison should let the proposed method also find the minimum number of pixels along with its optimal perturbation strength.
3. The baseline included in the experiments are pretty weak. For example, some baselines like iteratively conducting one-pixel adversarial attack should be considered rather than just comparing with a decision-based attack. Also, some dense attack could easily be changed to handle the sparse attack. For example, selecting the largest perturbation in the l-2 norm attack or using a relaxed l-1 attack would also be viable baselines.
4. Although the proposed method claims to largely improve the query efficiency, the number of query is still around in the same level with decision based attack. However, the score based attack should be much more query-efficient than decision-based attack in dense attack, where makes the score-based attack not attractive and impractical.

**Questions:**

Please refer to the weaknesses.

---

> ### Author Response · Authors · 2023-11-20
> **Response to Reviewer Xni7**
>
> Q1. Why only the optimal position?
>
> Indeed, an optimal search should also search for the optimal perturbation (value of pixels). Unfortunately, as we mention in __Section 1__, even without searching for the optimal perturbation, the problem is __*already NP-hard*__.
>
>   * So we need to find good enough *approximations*.  We propose not changing the colors to reduce the search space (see discussion in __Appendix H__). But, in spite of not changing the color or searching in the reduced space, our formulation outperforms the state-of-the-art.
>
>   * Overall, our *new insight* is that it is possible to find an “approximate” sparse adversarial example in a reduced search space with a very limited query budget (achieve query efficiency). Our results demonstrate that our approximations can find sparse adversarial examples with less queries than the current state-of-the-art method to pose a credible threat to deep learning models.
>
> Q2. Comparison with PGD_0 is not fair?
>
> Please recall, PGD$_0$ is a __white-box__ attack but ours is a __black-box__ attack. We used PGD$_0$ as an existing, popular, white-box $l_0$-attack baseline to put our black-box attack results into perspective and not for a 1-1 comparison.
>
>
> Q3. Some dense attack could easily be changed to handle the sparse attack? relaxed l-1 attack? One pixel attack adaptations?
>
> __*We considered these alternatives and generated results already*__.
> But, we did not attempt the idea to adapt single-pixel attacks. We have done this now added new results.
>
>
> Our approach outperformed all __*dense attack adaptations*__, as discussed in __Section 4__ and __Appendix D__.
>  - see __*Adapted l0 Attacks (White-box)*__
>  - see __*Adapted l0 Attacks (Decision-based)*__ using state-of-the-art query efficient attacks
>  - More importantly, recall the reason for __*why dense attack adaptations continue to perform poorly*__
>       - Because, to the best of our knowledge, there is no effective projection method to identify the pixels that can satisfy the L0 projection constraint (see __Section 4.2__, __Appendix D__ and study [1]).
>       - To demonstrate that, we compared our method with PGD$_0$ which is an adapted-$l_0$ attack in white-box settings and expected to be better than other adapted-$l_0$ attacks in black-box settings.
>
> Why did we not compare with relaxed-$l_1$ attack?
>   - The approaches [2,3] achieved great results but it did *not* perform well when compared with Sparse-RS as shown in [1].
>   - But we *outperformed* Sparse-RS (the current state-of-the art black box method), therefore there was no need to compare with [2,3].
>   - Actually, while developing BruSLeAttack, we also explored this approach but its performance was extremely poor in terms of needed queries. Using an $l_1$ constraint can lead to a three fold increase in the search space because now we need to search for  color values of each channel of each pixel as opposed to just limiting the number of manipulated pixels ($l_0$ constraint). So, we directly solved the problem in Equation (1) through $l_0$ constraint.
>
> Following your interesting idea, we now __*added a new result with a One-Pixel Attack*__
>
>   - We conducted an experiment with 1000 correctly classified images on CIFAR10 in untargeted settings (notably the easier attack, compared to targeted settings) using ResNet18  These images are evenly distributed across 10 different classes.
>   - We compare ASR between our attack and One-Pixel at different budgets e.g. one, three and five perturbed pixels. For One-pixel attack, we used the default setting with 1000 queries. To be fair, we set the same query limits for our attack. We adopted the implementation of One-Pixel at `https://github.com/Harry24k/adversarial-attacks-pytorch`.
> - The results in Table 1 show that our attack outperforms the One-Pixel attack across one, three and five perturbed pixels, even under the easier, untargeted attack setting.
>
> __Table 1__: ASR comparison (higher is better) between One-Pixel and BruSLeAttack against ResNet18 on CIFAR-10.
> |Perturbed Pixels|One-Pixels|BruSLeAttack|
> |---|---|---|
> |1 pixel|19.5%|27.9%|
> |3 pixels|41.9%|69.9%|
> |5 pixels|62.3%|86.4%|

---

> ### Author Response · Authors · 2023-11-20
> **Response to Reviewer Xni7**
>
> Q4. "Number of query is still around in the same level with decision based attack".
>
> *__We respectfully disagree__*. Our quantitative results contradict this statement.
>
> Our score-based attack is significantly more query efficient than the decision-based attacks in sparse settings, especially in the more practical, high-resolution ImageNet (224x224) task.
>
> In the high-resolution ImageNet task:
> - __*Figure 5*__ results clearly demonstrate that our algorithm does significantly better.
>   - Consider, even at 0.2% of sparsity (around 100 perturbed pixels over from 50,000 in ImageNet), our attack with a budget of 10K queries can achieve around 45% ASR while ASR obtained by SparseEvo (the current state-of-the-art score-based attack) even with __double the query__ budget (20K), is only 2%.
>   - Consider, at 0.4% sparsity (200 pixels), our attack with a 10K query budget reaches more than 80% ASR, while SparseEvo with a double query budget only obtains about 10% ASR.
> - As another baseline, consider 0.4% sparsity (200 pixels) result in __*Figure 4*__ but now with only a query budget of 2K, our attack is able to obtain around  40% ASR. In comparison, Figure 5 shows that SparseEvo with 20K queries, only obtains 10% ASR. Notably, our attack used only 1/10 the query budget but obtained four times higher ASR than SparseEvo, a state-of-the-art decision-based attack.
>
> In the low-resolution datasets (with an exhaustive testing protocol over the test set)
> - In CIFAR-10 (32x32) or STL-10 (96x96), we still observe the same results and confirm our observation on ImageNet.
> - However, as the search space is lower in low-resolution datasets, the outperformance of our attack may not be as significant as observed on the high-resolution dataset.
> - Actually, in practice, we can see that real-world Machine Learning as a Service offerings are for even higher resolution images than ImageNet e.g. 640 x 480 [1] which is nearly __*six*__ times the resolution on ImageNet. Therefore, being able to scale to high-resolution images makes our method more *relevant*, *practical* and *credible threat*.
>
> *References*
>
> [1] F Croce, M Andriushchenko, N D Singh, N Flammarion, and M Hein. Sparse-RS: A Versatile Framework for Query-Efficient Sparse Black-Box Adversarial Attacks. Association for the Advancement of Artificial Intelligence (AAAI), 2022.
>
> [2] A. Modas, S.-M. Moosavi-Dezfooli, and P. Frossard. Sparsefool: a few pixels make a big 369 difference. Computer Vision and Pattern Recognition (CVPR), 2019.
>
> [3] L. Schott, J. Rauber, M. Bethge, and W. Brendel. Towards the first adversarially robust neural network model on mnist. International Conference on Learning Recognition(ICLR), 2019
>
> [4] https://cloud.google.com/vision/docs/supported-files

---

### Meta-Review · Area_Chair_vRz1 · 2023-12-10

**Metareview:**

The authors propose a score-based, black-box $\ell_0$-attack based on a Bayesian framework. In the experiments they show that they consistently outperform Sparse-RS, the currently best score-based $\ell_0$-attack. The reviewers appreciate the extensive experiments on various models and datasets (robust and non-robust models). Additionally, the authors show that their attack works also better than Sparse-RS on a real API.

Strengths:
- very good and extensive experimental results where they consistently outperform Sparse-RS
- they outperform PGD_0 (a white box-attack), even though this has already been shown by Sparse-RS

Weaknesses:
- the scope of the paper is limited (only $\ell_0$ and here seemingly only the pixel-based threat model and not additionally the channel-based threat model even though the adaptation should be trivial)
- the motivation and presentation of the Bayesian framework lacks clarity
- the authors make a big point out of the reduction of the search space by sampling from a synthetic image and then just searching for the pixel positions where they replace the color of the original image with the one of the synthetic image. I think an important ablation is missing (which would show that the Bayesian framework works indeed better than the search in Sparse-RS): replace in Sparse-RS the update step by fixing the colors to be changed to the synthetic image. If then Brusleattack is still better, that would be a clear argument that it is indeed the better framework

Some of the reviewers have argued for rejection but their points were mainly invalid, others argued for weak acceptance. The scope of the paper is limited but the results compared to Sparse-RS (which however considered also other sparse threat models) are consistently very good. I strongly encourage the authors to further work on the clarity and motivation of the paper and include the suggested ablation of Sparse-RS in the final version of their paper.

**Justification For Why Not Higher Score:**

limited scope of the paper

**Justification For Why Not Lower Score:**

strong experimental results compared to the score-based black-box attack Sparse-RS which seems to be SOTA for $\ell_0$-attacks (also outperforming white-box attacks)

---

### Decision · Program_Chairs · 2024-01-16

Accept (poster)